# ACTIONS SPEAK LOUDER THAN PROMPTS: A LARGE-SCALE STUDY OF LLMS FOR GRAPH INFERENCE

**Ben Finkelshtein** *
University of Oxford

**Silviu Cucerzan**
Microsoft Research

**Sujay Kumar Jauhar**
Microsoft Research

**Ryen White**
Microsoft Research

## ABSTRACT

Large language models (LLMs) are increasingly used for text-rich graph machine learning tasks such as node classification in high-impact domains like fraud detection and recommendation systems. Yet, despite a surge of interest, the field lacks a principled understanding of the capabilities of LLMs in their interaction with graph data. In this work, we conduct a large-scale, controlled evaluation across several key axes of variability to systematically assess the strengths and weaknesses of LLM-based graph reasoning methods in text-based applications. The axes include the LLM-graph interaction mode, comparing prompting, tool-use, and code generation; dataset domains, spanning citation, web-link, e-commerce, and social networks; structural regimes contrasting homophilic and heterophilic graphs; feature characteristics involving both short- and long-text node attributes; and model configurations with varying LLM sizes and reasoning capabilities. We further analyze dependencies by methodically truncating features, deleting edges, and removing labels to quantify reliance on input types. Our findings provide practical and actionable guidance. (1) LLMs as code generators achieve the strongest overall performance on graph data, with especially large gains on long-text or high-degree graphs where prompting quickly exceeds the token budget. (2) All interaction strategies remain effective on heterophilic graphs, challenging the assumption that LLM-based methods collapse under low homophily. (3) Code generation is able to flexibly adapt its reliance between structure, features, or labels to leverage the most informative input type. Together, these findings provide a comprehensive view of the strengths and limitations of current LLM-graph interaction modes and highlight key design principles for future approaches.

## 1 INTRODUCTION

Large language models (LLMs) have rapidly evolved into versatile problem solvers with strong in-context learning, reasoning, and tool-use abilities (Brown et al., 2020; Wei et al., 2022). Their capabilities extend across natural language (Raffel et al., 2020), code completion and synthesis (Chen et al., 2021a), and cross-modal applications (Liu et al., 2023). Much like in other domains, recent work in graph machine learning has explored leveraging LLMs for tasks such as node classification (Fatemi et al., 2024; Wu et al., 2025), graph property prediction (Guo et al., 2023) and knowledge graph reasoning (Zhu et al., 2023), with node classification emerging as a dominant task.

This recent wave of interest stems from many high-impact node classification applications that are inherently text-rich and well suited to the strengths of LLMs. These include information retrieval (Su et al., 2024), fraud detection (Yang et al., 2025), and recommendation systems (Robinson et al., 2024). For instance, in fraud detection, accounts are nodes, transactions are edges, and the goal is to use textual and relational metadata to assign risk labels to nodes early to prevent financial losses.

Consequently, LLMs have emerged as a viable alternative to the dominant paradigm for graph understanding, Graph Neural Networks (GNNs) (Kipf & Welling, 2017a; Veličković et al., 2018; Finkelshtein et al., 2024), and exhibit competitive performance on text-rich graphs (Ye et al., 2024). While GNNs are typically trained per task and dataset and do not transfer across domains or label spaces (Finkelshtein et al., 2025), a key advantage of LLMs is their broad world-knowledge (Roberts

---

*Work performed while at Microsoft Research.

et al., 2020), which can benefit long-text graph datasets, such as e-commerce, web-link, and social networks (Shchur et al., 2019; Hu et al., 2020; Pei et al., 2020; Mernyei & Cangea, 2020; Hamilton et al., 2017). Furthermore, LLMs have many ways to process and reason over graph information: through linearization of text and prompt augmentation; to specialized tool usage for querying the underlying graph; to generating arbitrary code that operates over the graph. We refer to these different approaches as LLM-graph interaction strategies or modes.

However, despite rapid adoption of LLMs in graph understanding, and node classification in particular, most prior work targets performance for specific domains, graphs, or tasks. As a result, the field currently lacks a principled understanding of the capabilities of LLMs in ther interactions with graph information, and learnings that practitioners can leverage when integrating them into their scenarios.

This principled understanding is especially important, since there are many axes of variability – and blindly applying LLMs to graphs risks sub-optimal or even detrimental outcomes. Thus, in this paper we conduct a comprehensive, controlled, large-scale evaluation that factorizes these key axes: (1) the **LLM-graph interaction mode**, comparing four prompting variants, two ReAct-style tool-using variants (Yao et al., 2023), and a programmatic Graph-as-Code medium; (2) **dataset domains**, spanning citation, web-link, e-commerce, and social networks; (3) **structural regimes**, including homophilic and heterophilic graphs [1]; (4) **feature characteristics**, comparing short- and long-text attributes; (5) **model scale**, ranging from smaller to larger LLMs (across both open- and closed-source families) and (6) **reasoning capabilities**, contrasting reasoning and non-reasoning variants of LLMs.

Furthermore, to shed light on the inner workings of LLM-based approaches, we move beyond reporting overall accuracy and deepen our analysis by probing their reliance on features, structure, and labels at inference time. Specifically, we independently truncate textual features, remove known labels, and delete edges, producing 2D accuracy heatmaps that reveal each interaction mode's information dependencies. By isolating the contributions of features, labels, and structure, practitioners can identify which mode exploits particular types of information most effectively, thereby guiding the choice of interaction mode that best matches their application's characteristics (e.g., feature length, homophily) rather than relying on opaque, one-size-fits-all solutions.

**Findings.** Our evaluation yields key insights and guidelines in applying LLMs to graph data:

- **Graph-as-Code achieves the strongest overall performance**, with especially large gains on long-text or high-degree graphs where prompting quickly exhausts the token budget.
- **All LLM-graph interaction modes are effective on heterophilic graphs**, challenging the common assumption that they collapse under low homophily (Huang et al., 2024a).
- **Graph-as-Code is able to flexibly shift its reliance** between structure, features, or labels, leveraging the most informative input type.

Experiments on LLM size and reasoning capabilities can be found in Sections B.1 and B.2.

## 2  RELATED WORK

**Textualization and prompting of graphs.** Motivated by LLMs' cross-domain performance, early work encodes graphs as text for LLMs, benchmarking varying encoding styles such as adjacency lists, edge lists, shortest-path descriptions, and narrative-style encodings (Fatemi et al., 2024). Subsequent studies employ these textualizations to evaluate LLMs on node classification using prompting setups (Huang et al., 2024a; Wang et al., 2024; Li et al., 2024b; Dai et al., 2025; Ye et al., 2024; Guan et al., 2025). They find that carefully designed prompts can allow LLMs to compete with GNNs (Ye et al., 2024), that performance often hinges on neighborhood homophily (Huang et al., 2024a), and that their abilities remain brittle and sensitive to input data and formatting (Wang et al., 2024). More mechanistic analyses of attention patterns further suggest that LLMs may mirror prompt format rather than execute explicit graph computation (Guan et al., 2025).

Concurrently, alternatives to textualization have emerged, either by tokenizing each node based on local structure and features (Zhao et al., 2024) or by introducing learnable components that encode structure and features (Perozzi et al., 2024). Yet prompt-based approaches remain the default, with

---

[1]Homophily is the tendency of nodes to connect with others of the same class, whereas heterophily is the tendency to connect predominantly with nodes of different classes.

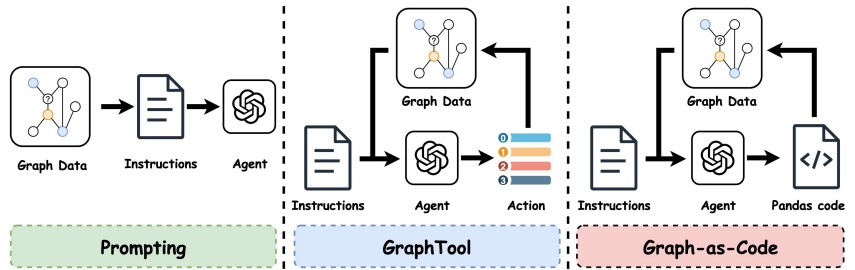

Figure 1: Illustration of the LLM-graph interaction strategies described in Section 3.1.

most recent work still relying on them (Wu et al., 2025; Tang et al., 2024; 2025; Guan et al., 2025). Our analysis contrasts prompting with tool-use and Graph-as-Code interaction modes, revealing competitive and often superior performance on graphs with large textual features or high-degree nodes. Contrary to conclusions drawn from prompting alone (Huang et al., 2024a), we also find all modes viable on heterophilous graphs, with the added advantage that Graph-as-Code shows reduced brittleness.

**Tool-calling for graph reasoning.** Recent advances in LLM orchestration have introduced tool-calling and ReAct-style paradigms, enabling LLMs to interface with external APIs or reasoning modules for enhanced problem-solving (Yao et al., 2023; Schick et al., 2023). These approaches interleave natural language reasoning with calls to task-specific tools, allowing LLMs to retrieve, manipulate, or compute over structured data. Variants such as Plan-and-Execute (Wang et al., 2023), Reflexion (Shinn et al., 2023), and Graph-of-Thought (Besta et al., 2024) have shown that LLMs can decompose complex tasks into sequences of actions and adapt plans via feedback. Tool-calling has been extensively benchmarked for real-world utility across domains – planning (Huang et al., 2024b), code API usage (Patil et al., 2024), mathematical reasoning (Gou et al., 2023), and multi-modal reasoning (Lu et al., 2023). In the context of graph data, LLMs have been combined with tool-calling to perform graph classification, knowledge graph reasoning and node classification, achieving high performance (Zhang, 2023; Edge et al., 2024). However, these works focus on specialized workflows or proof-of-concept demonstrations. We extend this line of research by systematically evaluating varying tool-calling paradigms for node classification across diverse datasets and graph regimes, highlighting their strengths, limitations and dependencies on features, labels and structure.

An additional related work discussion can be found in Section A.

## 3 AXES OF VARIABILITY

Our goal is to build a principled understanding of the capabilities of LLMs in processing graph information. In this section, we factorize key axes of variability to enable controlled comparisons that isolate the influence of each factor, and clarif their dependencies on graph features, structure, and labels.

**Notations.** Let $\mathcal{T} = \bigcup_{n \geq 0} \Sigma^n$ be the set of all finite token sequences over vocabulary $\Sigma$. We consider an unweighted graph $G = (V, E, \boldsymbol{X}, \boldsymbol{Y})$ with $N = |V|$ nodes, adjacency matrix $\boldsymbol{A} \in \{0, 1\}^{N \times N}$, node features $\boldsymbol{X} \in \mathcal{T}^N$, and labels $\boldsymbol{Y} \in \mathcal{C}^N$ for label set $\mathcal{C} \subseteq \mathcal{T}$ of size C. For any matrix $\boldsymbol{M} \in \mathbb{R}^{N \times D}$ and node subset $S \subseteq V$, $\boldsymbol{M}_S \in \mathbb{R}^{|S| \times D}$ denotes the submatrix of rows indexed by S.

**Node classification.** The task is to predict labels $\boldsymbol{Y}_Q \in \mathcal{C}^{|Q|}$ for a set of query nodes $Q \subset V$, given (i) known labels $\boldsymbol{Y}_K \in \mathcal{C}^{|K|}$ for a set of labeled nodes $K \subseteq V \setminus Q$, (ii) the graph structure (either $\boldsymbol{A}$ or $E$), and (iii) the textual features $\boldsymbol{X} \in \mathcal{T}^N$ of all nodes.

We model this task using the LLM-graph interaction models $\phi_{\text{prompt}}, \phi_{\text{tool}}, \phi_{\text{code}} : \mathcal{T} \times \mathcal{T}^N \times \{0, 1\}^{N \times N} \to \mathcal{T}$, corresponding to Prompting, GraphTool and Graph-as-Code, respectively. Each mode encodes the chat history in $\mathcal{T}$, node features in $\mathcal{T}^N$, and graph structure into a finite sequence, which is then processed by an $\text{LLM}_\theta : \mathcal{T} \to \mathcal{T}$ with parameters $\theta \in \mathbb{R}^D$. The output may provide predicted labels in $\mathcal{C} \subset \mathcal{T}$ or update the chat history for further interaction.

### 3.1 VARIABILITY OVER INTERACTION STRATEGIES AND MODEL CONFIGURATIONS

We study the following three LLM-graph interaction modes (Figure 1):

**(1) Prompting** $\phi_{\mathbf{prompt}}$**.** In this simple and widely-used mode (Fatemi et al., 2024; Huang et al., 2024a; Guan et al., 2025), the entire context for the model is constructed and issued to the model as a single-turn inference. The prompt (i) provides all classes (ii) presents the target node's textual description and known label (if available), and (iii) serializes the $k$-hop neighborhood grouped by hop distance, specifying for each encountered node its description and label (or *None* for held-out nodes). A complete prompt template for this mode is provided in Template 1.

The hop number is a hyperparameter controlling the degree of neighborhood information. We experiment with three variants: *0-hop prompt*, *1-hop prompt*, and *2-hop prompt*, corresponding to radii of 0, 1, and 2, respectively. To keep the context within a token budget for long-text datasets, we also experiment with an additional *budget prompt* variant, which caps the neighbors at each hop by subsampling.

**(2) GraphTool** $\phi_{\mathbf{tool}}$**.** Motivated by ReAct (Yao et al., 2023), we frame node classification as an iterative *think–act–observe* loop. At each step, the LLM reasons about what is known and what remains missing, then issues a single action from a fixed tool set. The environment executes the action on the graph and returns the result, which is appended to the interaction history. The process repeats until the LLM decides to terminate and predict a label. This ReAct-style interaction encourages planning and targeted retrieval of graph structure and text, reducing irrelevant exposure and token usage.

In our basic variation, *GraphTool*, the following actions are available: (0) The terminal action submits the final label. (1) A topology-only action retrieves the neighbors of a specified node, enabling exploration without consuming feature tokens. (2) A feature-only action returns the textual description of a specified node. (3) A label-only action reveals the label of the requested node if in the training set (and *None* otherwise), allowing the model to anchor reasoning on known examples while avoiding leakage on held-out nodes. We also introduce *GraphTool+*, which extends the base GraphTool variant with additional exact-$k$ hop retrieval actions: (4) retrieves the textual descriptions of all nodes exactly $k$ hops away from a specified node; (5) retrieves their labels (or *None* for held-out nodes). A complete prompt template for this mode is provided in Template 2.

**(3) Graph-as-Code** $\phi_{\mathbf{code}}$**.** Building on LLMs' strong code generation capabilities (Chen et al., 2021a; Liu et al., 2025), we extend the ReAct paradigm beyond a fixed, predefined action set. In this mode, the graph data is represented as a typed table indexed by `node_id` with columns `features` (text), `neighbors` (list of node IDs), and `label` (integer or *None*). The LLM generates, executes, then reasons over the outputs of compact programs in an iterative fashion. The process repeats until the LLM decides to terminate and predict a label. This code-native mode enables compositional access to structure and features and can collapse multi-step tool sequences into a single query, improving step and token efficiency while remaining transparent and auditable. A complete prompt template for this mode is provided in Template 3.

**Variability across LLM sizes and reasoning.** We evaluate models from small (`Llama`) to large (`GPT-5`), including reasoning and non-reasoning variants of `Phi-4` and `Qwen`, to assess the impact of scale and reasoning. `o4-mini` is our primary model; additional results are in the Appendix.

## 3.2 VARIABILITY OVER DATASET DOMAINS, HOMOPHILY LEVELS AND TEXT LENGTHS

**Dataset domains.** We evaluate LLM performance across diverse graph domains, such as citation network datasets *cora*, *pubmed*, and *arxiv*, where nodes are papers with titles as features (Huang et al., 2024a); e-commerce graph datasets *products*, *computers* and *photo* (Huang et al., 2024a; Wu et al., 2025), where nodes are items with product title, descriptions or reviews; web-link network datasets *cornell*, *texas*, *washington*, *wisconsin*, and *wiki-cs* (Wu et al., 2025), where nodes are webpages described by their page-level text; social network datasets *reddit* and *instagram* with user profiles and comment snippets (Wu et al., 2025). This domain variability enables a comprehensive assessment of LLM generalization and adaptation to different graph types.

**Graph structure regimes.** In homophilic graphs such as citation networks, e-commerce and social networks, local label information is highly important for correct prediction. Conversely, heterophilic graphs such as web-link networks, challenge models to rely less on simple local label information and more on node features and graph structure. By evaluating performance across these regimes, we aim to uncover the varying dependencies of LLMs on graph features, structure, and labels.

**Textual feature characteristics.** The datasets also vary in the richness and complexity of node textual features. Short-text datasets, such as *cora*, *pubmed*, *arxiv* and *products* provide only titles or product

Table 1: Accuracy of baselines and LLM-graph interaction modes Prompting, GraphTool, and Graph-as-Code on short-text homophilic datasets. Best per-dataset results are **bold**, runner-up underlined.

|  | cora | pubmed | arxiv | products |
|---|---|---|---|---|
| # Classes | 7 | 3 | 40 | 47 |
| Avg. text length | 66.13 | 110.47 | 68.99 | 54.08 |
| Hom. (%) | 82.52 | 79.24 | 65.53 | 63.84 |
| Avg. degree | 4.92 | 6.30 | 13.64 | 61.37 |
| Random | 14.13±1.06 | 33.10±1.19 | 2.44±0.23 | 2.33±0.25 |
| Majority label | 29.00±0.89 | 41.90±4.08 | 5.90±1.39 | 26.10±2.49 |
| Label propagation | 76.61±1.94 | 80.80±2.93 | 68.00±1.66 | 70.40±1.64 |
| 0-hop prompt | 64.17±0.68 | 89.20±1.89 | 68.10±3.11 | 70.00±5.82 |
| 1-hop prompt | 81.92±1.86 | 91.30±2.02 | 73.80 ±1.92 | 82.20±3.98 |
| 2-hop prompt | 83.43±2.25 | 91.80±2.17 | 74.30±2.53 | TokenLimit |
| GraphTool | 74.02±1.18 | 89.50±2.32 | 67.50±5.50 | 75.30±3.06 |
| GraphTool+ | 81.40±3.08 | **91.90**±2.16 | 73.30±2.86 | 78.50±3.43 |
| Graph-as-Code | **85.16**±1.47 | 89.90±1.85 | **74.40**±3.02 | **82.70**±2.66 |

names, offering limited semantic signal for classification. In contrast, long-text datasets such as *computers*, *photo*, *reddit*, *instagram* and *wiki-cs* include detailed descriptions or user profiles, presenting both opportunities for deeper reasoning and challenges for efficient context processing by LLMs.

# 4 EXPERIMENTS ACROSS AXES OF VARIABILITY

We evaluate LLMs across multiple axes of variability. We organize this section into three parts: short-text homophilic datasets (section 4.1), heterophilic datasets (section 4.2), and long-text homophilic datasets (section 4.3). In each setting, we introduce additional baselines to contextualize performance.

**Baselines.** We compare against several baselines: **Random**, which predicts labels uniformly at random, serving as a natural lower bound; **Majority Label**, which assigns the most frequent label from the training and validation sets to all test nodes; and the classic **Label Propagation** (LP) algorithm. In LP, node labels are represented as one-hot vectors $Y \in \{0, 1\}^{N \times C}$, unknown labels are initialized as zero vectors. The (random-walk) *normalized adjacency matrix* is defined as $\hat{A} = D^{-1}A$, where $D$ is a diagonal degree matrix $D = \text{diag}(d_1, \ldots, d_n)$ with $d_i = \sum_{j=1}^{n} A_{ij}$ denoting the degree of node $i$. Predictions $\hat{Y} \in \mathbb{R}^{N \times C}$ are computed by $\hat{Y} = \hat{A}^{\ell}Y$, where $\ell = 10$ is the number of steps, and each node is assigned the label with the highest score.

## 4.1 SHORT-TEXT HOMOPHILIC DATASETS

**Finding 1.** *Prompting and Graph-as-Code are closely competitive on short-text homophilic datasets.*

Table 1 reaffirms prior work by showing that all LLM-based approaches substantially outperform trivial baselines such as random guessing and majority label assignment. This confirms LLMs leverage both textual node features and graph structure for classification in homophilic regimes. Furthermore, within the Prompting interaction mode, accuracy increases with the inclusion of neighborhood context, moving from self to 1-hop and 2-hop prompt variants, consistent with established findings Huang et al. (2024a); Wu et al. (2025). However, *on graphs with high average degree, context token limits are reached quickly*, restricting possible gains from additional neighborhood information.

**Finding 2.** *In ReAct-based methods, more flexible variants perform better on homophilic datasets.*

The ReAct-based interaction strategies exhibits a clear trend from GraphTool to GraphTool+ to Graph-as-Code. As the LLM is given greater agency and adaptivity in interacting with the graph – moving from fixed tool invocation (GraphTool), to enhanced retrieval options (GraphTool+), and finally to the fully programmatic Graph-as-Code—classification, accuracy improves (Finding 2). We believe that adaptivity is valuable in homophilic settings, as local neighborhood labels are highly informative but the optimal aggregation strategy may also depend on node degree and graph topology. Thus, increased agency empowers the LLM to tailor its reasoning and retrieval to the specific structure of each instance, resulting in stronger overall performance.

Table 2: Accuracy of baselines and LLM-graph interaction modes Prompting, GraphTool, and Graph-as-Code on heterophilic datasets. Best per-dataset results are **bold**, runner-up underlined.

|  | cornell | texas | washington | wisconsin |
|---|---|---|---|---|
| # Classes | 5 | 5 | 5 | 5 |
| Avg. text length | 2039.69 | 2427.40 | 1597.53 | 2109.54 |
| Hom. (%) | 11.55 | 6.69 | 17.07 | 16.27 |
| Avg. degree | 1.53 | 1.66 | 1.72 | 1.89 |
| Random | 21.74±3.19 | 8.40±1.56 | 20.43±2.87 | 18.11±3.62 |
| Majority label | 42.43±1.56 | 58.70±1.40 | 45.94±3.64 | 44.15±2.20 |
| Label propagation | 41.74±1.06 | **78.90**±1.67 | 15.07±4.21 | 14.21±2.69 |
| 0-hop prompt | 81.57±1.80 | 53.20±3.19 | 80.14±2.54 | 84.78±2.86 |
| 1-hop prompt | 81.39±0.99 | 71.40±2.07 | 81.74±1.80 | 88.81±1.43 |
| 2-hop prompt | 84.17±1.43 | TokenLimit | **84.35**±1.67 | **91.45**±1.87 |
| GraphTool | 91.30±2.46 | 59.60±2.38 | 80.14±1.32 | 87.04±1.58 |
| GraphTool+ | 91.13±2.97 | 63.70±2.36 | 80.41±0.94 | 87.42±1.60 |
| Graph-as-Code | **92.70**±2.35 | 73.60±3.78 | 81.96±2.92 | 89.17±2.69 |

Table 3: Accuracy of baselines and LLM-graph interaction modes Prompting, GraphTool, and Graph-as-Code on long-text homophilic datasets. Best per-dataset results are **bold**, runner-up underlined.

|  | citeseer | reddit | computer | photo | instagram | wiki-cs |
|---|---|---|---|---|---|---|
| # Classes | 6 | 2 | 10 | 12 | 2 | 10 |
| Avg. text length | 1018.97 | 761.82 | 792.77 | 797.82 | 509.64 | 3215.56 |
| Hom. (%) | 72.93 | 55.52 | 85.28 | 78.50 | 63.35 | 68.67 |
| Avg. degree | 1.34 | 5.93 | 8.27 | 10.36 | 12.70 | 18.45 |
| Random | 16.80±2.05 | 51.60±2.97 | 10.00±2.21 | 8.20±3.03 | 50.30±3.07 | 9.90±2.13 |
| Majority label | 21.90±2.10 | 52.80±2.97 | 24.20±2.89 | 42.30±2.25 | **65.10**±2.90 | 21.50±3.89 |
| Label propagation | 37.30±5.03 | 40.50±1.66 | 74.70±1.60 | 75.30±0.91 | 52.00±2.98 | 71.90±2.19 |
| 0-hop prompt | 68.20±2.77 | 47.90±3.86 | 65.50±3.32 | 69.80±3.15 | 48.00±3.66 | 74.00±3.10 |
| 1-hop prompt | 68.30±2.59 | 59.30±4.12 | 86.10±2.59 | 85.80±1.92 | 56.10±2.30 | TokenLimit |
| 2-hop prompt | 69.40±2.10 | TokenLimit | TokenLimit | TokenLimit | TokenLimit | TokenLimit |
| 2-hop budget prompt | 70.20±2.83 | 54.40±3.85 | 86.00±2.41 | 85.60±1.63 | 54.50±4.45 | 80.80±3.48 |
| GraphTool | 68.30±1.15 | 56.25±1.84 | 80.80±4.61 | 77.00±2.32 | 47.80±4.09 | 76.27±4.18 |
| GraphTool+ | 68.70±2.31 | **61.80**±1.15 | 83.10±2.63 | 81.30±0.97 | 48.20±2.92 | 80.50±3.10 |
| Graph-as-Code | **71.80**±2.22 | 61.60±2.36 | **86.20**±3.55 | **86.40**±2.65 | 56.40±2.56 | **82.20**±3.63 |

## 4.2 HETEROPHILIC DATASETS

**Finding 3.** *All LLM-graph interaction modes are effective on heterophilic datasets.*

Table 2 challenges common assumptions and prior work (Huang et al., 2024a) which suggests LLMs struggle on heterophilic graphs. In this setting, local label information can be non-predictive or even misleading, making it difficult for LLMs to rely on neighborhood cues for accurate classification. Despite low levels of homophily, all LLM-graph interaction modes ($\phi_{\text{prompt}}, \phi_{\text{tool}}, \phi_{\text{code}}$) achieve strong accuracy (Finding 3), consistently outperforming classic baselines such as majority labeling and label propagation. This demonstrates that LLMs can exploit non-local or feature-based cues for classification, rather than relying solely on simple neighborhood voting heuristics. Here, the *context window token limit is reached again, primarily due to long textual features*, rather than graph degree, which constrains the amount of context that can be included in prompts. Nevertheless, providing more context in prompting proves beneficial, contrary to popular belief, and similar to the homophilic setting.

Furthermore, ReAct-based variants exhibit a clear advantage with increased agency and adaptivity in interacting with the graph – moving from fixed tool invocation (GraphTool), to enhanced retrieval options (GraphTool+), and finally to the fully programmatic Graph-as-Code, reinforcing Finding 2 in the heterophilic setting as well. The Graph-as-Code variant ($\phi_{\text{code}}$) in particular excels, likely due to its compositional access to features and structure, which is especially advantageous when neighborhood labels are diverse or uninformative.

## 4.3 LONG-TEXT DATASETS

**Finding 4.** *Graph-as-Code significantly outperforms Prompting and GraphTool on long-text datasets.*

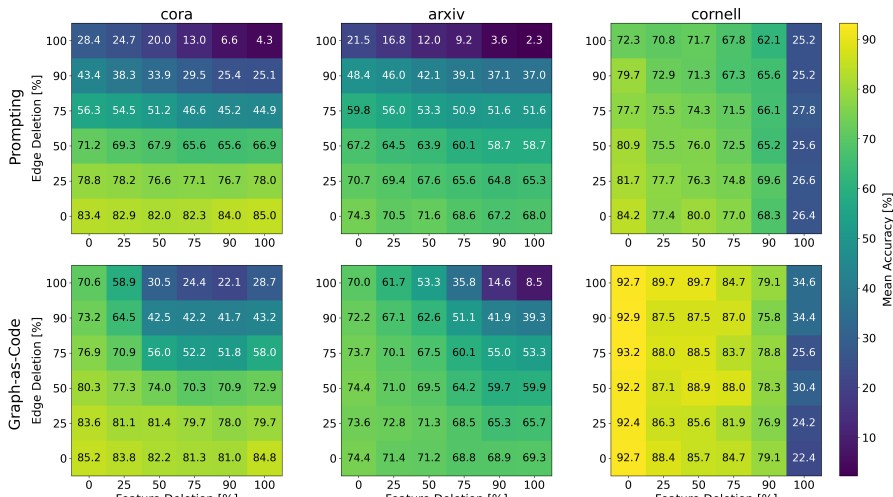

Figure 2: Accuracy of 2-hop prompting and Graph-as-Code on the cora, arxiv, and cornell datasets under varying ratios of randomly removed edges and truncated text features.

Table 3 suggests of a performance gap, with Prompting ($\phi_{\text{prompt}}$) performing worst and Graph-as-Code best (Finding 4). In this setting, Prompting is constrained by node feature length and neighborhood size, both quickly exhausting the model's token budget. Notably, similar token-limit issues appeared in previous sections—for the *products* dataset in short-text homophilic benchmarks and for the *texas* and *washington* datasets in the heterophilic regime when features were long or neighborhoods large.

To mitigate these constraints, we introduce the 2-hop budget prompting variant, which caps the number of neighbors per node via sampling. While this adjustment helps avoid hitting the token limit and allows the model to reason over a sampled subset of context, Prompting still remains the least effective variant, with Graph-as-Code the most effective one (Finding 2). This is likely due to the noise and information loss introduced by sampling, which can obscure important neighborhood signals.

**These results demonstrate that Graph-as-Code ($\phi_{\text{code}}$) can offer substantial advantages for LLM-based node classification in dense or feature-rich graphs**, which represent a large proportion of real-world networks such as e-commerce and recommendation networks. Furthermore, while context windows are indeed expanding, prior work has shown that LLMs often struggle to effectively leverage long inputs, and that more context does not always translate to better reasoning (An et al., 2024; Li et al., 2024a). Thus, approaches like Graph-as-Code that restructure graph information to highlight salient structure and reduce redundancy remain crucial, even as model capacities grow.

## 5 EXPERIMENTS ON FEATURES, STRUCTURE, AND LABELS DEPENDENCIES

In this section, we examine whether the widely used Prompting ($\phi_{\text{prompt}}$) and the best performing Graph-as-Code ($\phi_{\text{code}}$) interaction modes rely similarly on node features, graph structure, and labels. We organize the analysis into two parts: (i) the effect (or dependency) of removing portions of node features and edges (section 5.1), and (ii) the effect of removing labels and edges (section 5.2). We visualize results with 2D heatmaps of accuracy against feature, edge, and label removal rates.

**Additional setup.** For each dataset, deletion rates, and LLM-graph interaction strategies, we predict the labels of $1,000$ randomly sampled test nodes per seed and average over five independent runs, reporting the mean as in previous experiments. However, in both Section 5.1 and Section 5.2 we run partial-deletion experiments: edges and labels are removed uniformly at random, while feature deletion is implemented by truncating each node's text to the fixed percentage of tokens. We adopt truncation as it provides a simple, model-agnostic, and reproducible way to scale the information available to the LLM, while avoiding additional assumptions about feature semantics that could bias results.

### 5.1 FEATURES VS. STRUCTURE DEPENDENCIES

**Finding 5.** *Prompting and Graph-as-Code exhibit comparable use of node features and structure.*

Figure 2 shows that, for all three datasets, the two panels have nearly identical characteristics, indicating that Prompting ($\phi_{\text{prompt}}$) and Graph-as-Code ($\phi_{\text{code}}$) share the same dependence on features and structure. On *cora* and *arxiv*, accuracy drops mainly with edge deletion, while on *cornell* it is driven primarily by feature deletion (Finding 5). This alignment arises from the inherent characteristics of the datasets. Both *cora* and *arxiv* are highly homophilic, meaning nodes of the same class are densely interconnected. In these settings, structural information, specifically the local label context provided by edges, has high impact on accuracy, so removing edges disrupts information flow and leads to a decrease in accuracy for both methods. Conversely, *cornell* is a heterophilic dataset, where nodes of different classes are more likely to be connected and the graph structure is less informative. Here, node features are more discriminative than the sparse and less meaningful edge connections, making feature deletion the dominant factor impacting performance.

**Finding 6.** *Graph-as-Code is more robust than Prompting to feature, structure and label deletion.*

Beyond the shared dependency, Graph-as-Code ($\phi_{\text{code}}$) consistently outperforms Prompting and is more resilient to perturbations (Finding 6). When structure is completely removed but features are intact, Graph-as-Code preserves high accuracy on all datasets, whereas Prompting collapses. This difference arises because Graph-as-Code can access feature and label information of other nodes even when edges are absent, whereas Prompting depends on edge connections to retrieve this information.

**Finding 7.** *When the prompt size reaches the token limit, the behavior of Graph-as-Code and Prompting diverges, with Graph-as-Code performing significantly better.*

Long-text homophilic graphs further challenge Finding 5 because 2-hop prompts are prone to hitting the token limit. Figure 3 shows that on *photo* we indeed observe a divergence in behavior between the two methods (Finding 7). This divergence on *photo* is expected given the dataset's characteristics. *Photo* is highly homophilic and contains nodes with rich, lengthy feature descriptions. In homophilic graphs, nodes are densely connected to others of the same class, so 2-hop prompts accumulate a substantial amount of feature text from numerous neighbors. This quickly exceeds the LLM's context window, leading to a significant drop in accuracy for Prompting. In contrast, Graph-as-Code ($\phi_{\text{code}}$) is designed to selectively retrieve and compose only the necessary structure and features for each query. This allows it to avoid exceeding the token limit and maintain high accuracy, even in the presence of long node descriptions and dense connectivity.

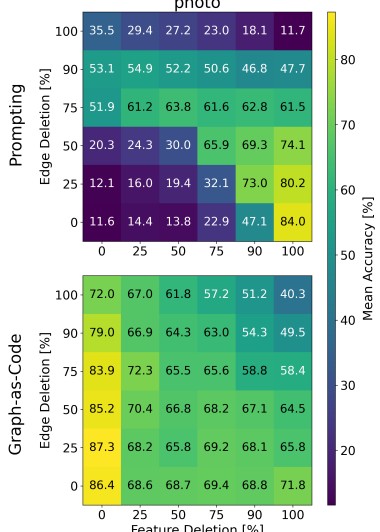

Figure 3: Accuracy of 2-hop prompting and Graph-as-Code on the photo dataset under varying ratios of randomly removed edges and truncated features.

This result ties back to our long-text homophilic experiments and findings in Section 4.3. There we observed a large gap in favor of Graph-as-Code ($\phi_{\text{code}}$); the *photo* ablation reveals the same characteristic: Prompting is fundamentally bottlenecked by the context window and can even benefit from discarding feature text to fit within it, whereas Graph-as-Code ($\phi_{\text{code}}$) retrieves and composes the needed structure and features without exceeding the token budget. Thus, the two methods share the same dependence on features and structure when prompts fit the context window, but once the token limit becomes a limiting factor, their behaviors diverge and Graph-as-Code proves superior. Practitioners should therefore assess the graph density and average feature length before choosing an LLM-graph interaction mode, prioritizing adaptive methods such as Graph-as-Code for cases with high density or long-text features.

## 5.2 LABELS VS. STRUCTURE DEPENDENCIES

**Finding 8.** *Prompting and Graph-as-Code exhibit different dependencies on labels and structure.*

Figure 4 demonstrates a stark contrast between the dependence patterns of Prompting ($\phi_{\text{prompt}}$) and Graph-as-Code ($\phi_{\text{code}}$) when subjected to edge and label deletion (Finding 8). This behavior is in contrast to the alignment on feature and structure dependencies observed in Section 5.1.

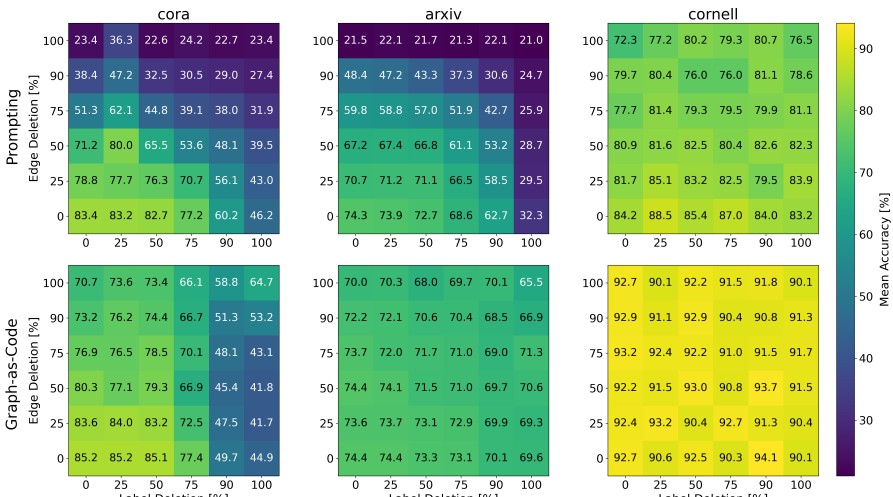

Figure 4: Accuracy of 2-hop prompting and Graph-as-Code on the cora, arxiv, and cornell datasets under varying ratios of randomly removed edges and known labels.

**Finding 9.** *Graph-as-Code is able to flexibly shift its reliance between structure, features, and labels to leverage the most informative input type.*

For Prompting ($\phi_{\text{prompt}}$), the results across all datasets reveal that accuracy degrades rapidly along both axes, confirming that Prompting requires both structure and labels to perform reasonably. Graph-as-Code ($\phi_{\text{code}}$) by comparison, displays a notably different pattern. Its accuracy remains nearly constant as edges are deleted, as long as either features or labels are present. This might suggest that Graph-as-Code ignores structure entirely; however, as previously observed in Section 5.1, structural information becomes crucial only when features are truncated. Thus, Graph-as-Code does not disregard structure, but instead leverages it only when it is more informative relative to other available signals.

This leads to a key insight: Graph-as-Code ($\phi_{\text{code}}$) can flexibly shift its reliance to the most informative input type, and is thus only vulnerable when multiple sources of information are heavily degraded (Finding 9). This adaptive behavior contrasts sharply with the brittleness of Prompting and re-emphasizes the robustness of Graph-as-Code (Finding 6).

## 6 CONCLUSIONS

In this work, we conducted the first comprehensive, controlled evaluation of LLMs for node classification across key axes of variability: LLM-graph interaction mode (Prompting, ReAct-style tool use, and Graph-as-Code), dataset domain (citation, web-link, e-commerce, social), structural regimes (homophilic vs. heterophilic), textual feature characteristics (short vs. long), model size (from small to large) and reasoning capabilities. Our large-scale study reveals that the Graph-as-Code method, which leverages LLMs' coding capabilities, achieves the strongest overall performance—especially on graphs with long textual features or high-degree nodes, where the widely-used prompting method quickly becomes infeasible due to context window limitations. We also find that all LLM-graph interaction methods remain effective on heterophilic graphs, challenging the commonly held assumption that LLM-based methods fail in low-homophily settings (Huang et al., 2024a).

Through a series of controlled dependency analyses, we independently truncate features, delete edges, and remove labels to quantify reliance on different input types. Experiments show that Graph-as-Code flexibly adapts its reliance to the most informative signal, be it structure, features, or labels.

Our findings provide actionable guidance for both practitioners and researchers: (1) Code generation is the preferred LLM-graph interaction mode, particularly as graphs grow in size and complexity; (2) LLMs remain effective on heterophilic graphs; (3) Graph-as-Code's adaptive reliance can be used to robustly handle noisy or partially missing data, where different input signals may be degraded.

## ETHICS STATEMENT

This work studies how large language models interact with graph-structured, text-rich data across prompting, tool-use, and code-generation modes. The work focuses on the node classification task in domains such as citation, web-link, e-commerce, and social networks. Potential benefits include safer fraud detection, improved recommendation, and better information retrieval; there are no potential risks. Our study uses established benchmark datasets and does not involve new human subjects data collection. Where social or user-generated content is present in benchmarks, we follow dataset licenses and use standard train/validation/test protocols; labels for held-out nodes are never revealed at inference time in our setups. We report results averaged over multiple seeds and include ablations to surface model dependencies on structure, features, and labels, which can help practitioners assess failure modes before deployment. We will comply with the ICLR Code of Ethics.

## REPRODUCIBILITY STATEMENT

To facilitate reproduction: (1) we specify all interaction templates used in our experiments (Prompting, GraphTool+, and Graph-as-Code) in the appendices, along with detailed task instructions and action formats. (2) We document data sources, domains, homophily levels, average degrees and text lengths. (3) We specify the model types used, along with their reasoning or non-reasoning variants. (4) We provide full detail over the evaluation procedures including per-seed sampling.

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

## A    ADDITIONAL RELATED WORKS

**LLMs coding capabilities.** Code generation has become a native capability of modern LLMs, with early systems demonstrating high-quality programs from natural language prompts (Chen et al., 2021b), even reaching competition-level performance (Li et al., 2022) and fueling the development of strong open-source models such as StarCoder (Li et al., 2023) and CodeGeeX (Zheng et al., 2023). Beyond pure code synthesis, program-aided approaches have leveraged code as a medium for reasoning—enabling LLMs to generate and execute short programs for mathematical problem solving (Gao et al., 2023), automate scientific workflows (Seo et al., 2025), and enhance reliability through self-debugging and iterative refinement (Chen et al., 2023). Recently, these capabilities have been extended to graph domains, where LLMs are prompted to generate code for solving simple graph problems such as cycle detection, connectivity analysis, and node degree estimation (Cai et al., 2024). We extend this further with Graph-as-Code: the LLM iteratively writes and executes concise programs over a standardized API to flexibly combine structural queries, label propagation, and textual feature processing. We then compare Graph-as-Code to prompting and tool-calling across datasets and asses its dependencies on the input feature, structure and labels.

**Hybrid LLM-GNN architectures.** Beyond textualization, recent works have proposed hybrid frameworks that integrate Graph Neural Networks (GNNs) or trainable projectors directly into the LLM architecture. Approaches closer to textualization, such as LLaGA (Chen et al., 2024), serialize each node's neighborhood into structure-aware sequences; however, instead of using raw text, these sequences are mapped via a trained linear projector into the LLM's input embedding space, enabling the processing of graph features as soft tokens. Other frameworks focus on explicit modality alignment. LangTopo (Guan et al., 2024) employs a GNN to map textual and spatial information into a discrete codebook of topological embeddings via Gumbel-softmax relaxation. It maximizes consistency between the LLM's natural language representations and these quantized codes, effectively transferring structural modeling capabilities to the LLM. Alternatively, cascaded architectures like UniGraph (He et al., 2025) directly fuse inputs by generating dense vector embeddings with a pre-trained GNN and concatenating them with the discrete token embeddings of natural language instructions. This combined sequence is fed into an LLM instruction-tuned to unify label spaces. While these hybrid methods rely on training auxiliary encoders or fine-tuning to fuse modalities, our work investigates the capabilities of frozen LLMs, evaluating how they can be orchestrated to reason over graphs at inference time through prompting, tools, and code generation.

## B    ADDITIONAL AXES OF VARIABILITY

In this section, we present experiments and findings across additional axes of variability. In Section B.1 and Section B.2, we analyze the impact of LLM size and reasoning, respectively, on accuracy across various graph domains, homophily levels, and feature text lengths.

### B.1    LLM SIZES

**Finding 10.** *Larger LLMs consistently deliver better performance.*

Table 4 demonstrate a clear and consistent trend from `Llama` to `o4-mini`, `DeepSeek R1`, and `GPT-5`: increasing LLM size yields improved node classification performance (Finding 10). This holds across all datasets and both interaction modes. This result is natural as larger LLMs possess greater capacity to model complex relationships and capture subtle dependencies between textual features and graph structure (Dai et al., 2025). As a results, the performance gap between Prompting and Graph-as-Code also persists at each model size, strengthening earlier findings from both short-text and long-text datasets that Graph-as-Code consistently provide an advantage (Findings 3 and 4).

We also note that the incremental improvements between `o4-mini`, `DeepSeek R1`, and `GPT-5` are less dramatic than the jump from `Llama`, suggesting diminishing returns at the upper end, but the trend is robust. This aligns with the broader literature on LLM scaling laws, where larger models better capture complex dependencies.

**Finding 11.** *Context window token limits consistently constrain the performance of Prompting, regardless of LLM size.*

Table 4: Accuracy of LLM-graph interaction modes Prompting and Graph-as-Code across LLMs of varying sizes.

| | Dataset | Llama | o4-mini | DeepSeek R1 | GPT-5 |
|---|---|---|---|---|---|
| | cora | 72.17±2.07 | 85.16±1.47 | 86.20±1.10 | 87.10±0.90 |
| | arxiv | 62.80±2.20 | 74.40±3.02 | 75.30±2.50 | 76.20±2.00 |
| | cornell | 80.05±1.90 | 84.17±1.43 | 85.50±1.20 | 86.20±1.00 |
| 2-hop prompt | texas | TokenLimit | TokenLimit | TokenLimit | TokenLimit |
| | citeseer | 14.10±4.55 | 69.40±2.10 | 71.00±1.90 | 72.20±1.70 |
| | photo | TokenLimit | TokenLimit | TokenLimit | TokenLimit |
| | wiki-cs | TokenLimit | TokenLimit | TokenLimit | TokenLimit |
| | **average** | 57.28±2.68 | 78.78±2.01 | 79.75±1.68 | 80.43±1.65 |
| | cora | 75.23±2.00 | 85.16±1.47 | 86.50±0.95 | 87.70±0.80 |
| | arxiv | 65.60±2.30 | 74.40±3.02 | 75.60±2.10 | 76.50±1.80 |
| | cornell | 86.00±2.10 | 92.70±2.35 | 93.10±2.00 | 93.80±1.80 |
| Graph-as-Code | texas | 68.00±2.50 | 73.60±3.78 | 75.10±2.60 | 76.00±2.40 |
| | citeseer | 16.30±4.00 | 71.80±2.22 | 72.80±1.95 | 73.60±1.85 |
| | photo | 83.69±2.59 | 86.40±2.65 | 87.00±2.50 | 87.60±2.20 |
| | wiki-cs | 79.76±2.94 | 82.20±3.63 | 83.10±3.10 | 83.80±2.90 |
| | **average** | 67.80±2.35 | 80.75±2.70 | 81.89±2.17 | 82.71±1.96 |

Table 5: Accuracy of LLM-graph interaction strategies Prompting and Graph-as-Code across LLMs with reasoning and non-reasoning modes.

| | Dataset | Phi-4 | | Qwen | |
|---|---|---|---|---|---|
| | | w.o. reasoning | reasoning | w.o. reasoning | reasoning |
| | cora | 80.16±1.60 | 87.38±1.32 | 80.72±1.57 | 88.05±1.25 |
| | arxiv | 69.40±2.80 | 75.83±2.32 | 70.10±2.70 | 77.11±2.10 |
| | cornell | 79.17±1.43 | 85.09±1.35 | 79.62±1.49 | 86.01±1.23 |
| 2-hop prompt | texas | TokenLimit | TokenLimit | TokenLimit | TokenLimit |
| | citeseer | 64.40±2.10 | 70.84±1.90 | 64.99±2.05 | 71.42±1.80 |
| | photo | TokenLimit | TokenLimit | TokenLimit | TokenLimit |
| | wiki-cs | TokenLimit | TokenLimit | TokenLimit | TokenLimit |
| | **average** | 73.28±1.98 | 79.29±1.72 | 73.36±1.95 | 80.15±1.60 |
| | cora | 81.08±1.47 | 89.12±1.10 | 81.67±1.52 | 89.83±1.08 |
| | arxiv | 71.40±3.00 | 78.54±2.40 | 72.01±2.95 | 79.21±2.30 |
| | cornell | 87.70±2.35 | 96.47±2.00 | 88.19±2.30 | 97.01±1.95 |
| Graph-as-Code | texas | 64.40±3.10 | 71.62±2.87 | 65.18±3.05 | 72.29±2.81 |
| | citeseer | 66.80±2.22 | 74.48±2.00 | 67.39±2.14 | 75.18±1.90 |
| | photo | 82.08±2.65 | 90.29±2.20 | 82.58±2.60 | 91.09±2.18 |
| | wiki-cs | 78.20±3.63 | 85.81±3.10 | 78.65±3.59 | 86.54±3.06 |
| | **average** | 75.38±2.42 | 83.19±2.10 | 75.67±2.39 | 83.45±2.04 |

A notable secondary observation is that token limits remain a bottleneck for Prompting in several datasets (*texas*, *photo*, *wiki-cs*), regardless of LLM size (Finding 11). This is expected as context window limitations are a fundamental property of LLM architectures – no matter how large the model, there is a maximum input length that cannot be exceeded. As neighborhood size or feature length increases, the input quickly outgrows this limit, preventing the LLM from accessing all relevant context. This limitation highlights a fundamental constraint of the Prompting strategy: as neighborhood or feature size grows, even large models cannot circumvent context window restrictions.

## B.2 Reasoning Capabilities

**Finding 12.** *Reasoning consistently improves performance.*

Table 5 reveals a consistent and substantial boost in accuracy when reasoning modes are enabled for both `Phi-4` and `Qwen` LLMs, regardless of the interaction strategy used. This pattern is expected, as reasoning has been shown to elevate LLM performance across various domains such as question answering, knowledge graph completion, and complex decision-making (Wei et al., 2022; Creswell & Shanahan, 2022; Zhou et al., 2022). It encourages the LLM to engage in step-by-step inference, explanation, and synthesis – capabilities that are crucial for effectively leveraging both node attributes and graph structure. For example, reasoning can help the LLM to better contextualize information from multi-hop neighborhoods and to integrate evidence from various sources, reducing the risk of overlooking important relationships or introducing spurious correlations.

## C Token and Latency Efficiency

In this section, we analyze the practical overhead of the evaluated LLM-graph interaction modes. Specifically, we assess token consumption in Section C.1 and computational latency in Section C.2 to highlight the real-world deployment trade-offs between the different interaction modes.

## C.1 Token Consumption

To assess the efficiency and scalability of the evaluated interaction strategies, we report the average token consumption per query on long-text datasets in Table 6. These metrics illustrate why context window limits are a critical bottleneck for prompting variants.

Table 6: Average token counts per query for each LLM-graph interaction modes Prompting, Graph-Tool and Graph-as-Code over long-text datasets.

| Dataset | 0-hop prompt | 1-hop prompt | 2-hop prompt | GraphTool | Graph-as-Code |
|---------|--------------|--------------|--------------|-----------|---------------|
| reddit  | 1,762        | 10,300       | 42,100       | 16,100    | 14,900        |
| photo   | 3,000        | 36,100       | 104,700      | 43,300    | 41,500        |
| wiki-cs | 30,200       | 202,500      | 1,157,000    | 61,700    | 57,100        |

As demonstrated in Table 6,, the total token count increases significantly from *reddit* to *photo*, and again to *wiki-cs*. This trend is expected, as it mirrors the increase in the average text length and average node degree for these datasets. Consequently, 1-hop and 2-hop prompts quickly exceed the context window of $200,000$ tokens for our default `o4-mini` model forcing truncation. In contrast, **GraphTool and Graph-as-Code use much smaller token counts, as they only retrieve specific pieces of information** (e.g., a list of neighbor IDs or a single node's features) and build context iteratively.

## C.2 Computational Efficiency

We assess the computational efficiency of the evaluated interaction strategies for practical use by measuring the average wall-clock time per query on the long-text datasets *reddit*, *photo*, and *wiki-cs*. This latency is measured from the submission of the initial prompt until the LLM yields its final prediction, with results averaged across all evaluated nodes.

Table 7: Average latency (wall-clock time in seconds) per query across for each LLM-graph interaction modes over long-text datasets.

| Dataset | 2-hop prompt | GraphTool | Graph-as-Code |
|---------|--------------|-----------|---------------|
| reddit  | 41           | 45        | 43            |
| photo   | 125          | 131       | 128           |
| wiki-cs | TokenLimit   | 159       | 155           |

The results in Table 7 indicate that the primary driver of latency is the dataset's characteristics (e.g., text length, graph density) rather than the interaction mode itself. This trend is clear across the datasets: *wiki-cs*, which has the longest text features and highest average degree, shows the highest latency. This is followed by *photo*, which is also text-rich and has a relatively high degree. *Reddit*, in contrast, has the shortest text and lowest average degree of the three, which results in the lowest latency. This is because processing longer text and larger, more complex neighborhoods simply requires more overall computation from the LLM, regardless of the method.

The modes are all dominated by the LLM's inference time, which is why their latencies are so comparable. While GraphTool+ and Graph-as-Code involve multiple iterative calls, each call is very "lightweight" (e.g., retrieving only neighbor IDs or a single feature). In contrast, Prompting makes a single, "heavy" call with a massive context. In practice, these different approaches balance out: the total computational load of many small calls becomes comparable to that of one large, token-heavy call.

## D    ADDITIONAL EXPERIMENTS

In this section, we explore the sensitivity of Prompting to neighborhood depth (Section D.1), asses the impact of code-generation priors on the varying interaction modes (Section D.2), extend our evaluation to algorithmic graph reasoning tasks (Section D.3), and compare our zero-shot interaction modes against supervised baseline architectures (Section D.4).

### D.1    THE IMPACT OF THE NUMBER OF HOPS ON PROMPTING

A potential consideration when comparing Prompting ($\phi_{\text{prompt}}$) to Graph-as-Code ($\phi_{\text{code}}$) is the number of hops accessed. Prompting is fundamentally constrained by token limits, which upper-bounds the number of hops that can be included in the context window. In contrast, ReAct-based methods can dynamically query the graph, adjusting the exploration depth on a per-node basis to effectively manage the token budget. This varying resource exposure is not a confounding variable, but rather a fundamental, defining property that differentiates these interaction modes in practice. The strict context boundary is a primary real-world limitation of Prompting, while the capacity for efficient programmatic exploration defines Graph-as-Code.

Nevertheless, to provide a rigorous comparison, we introduce an iterative *k-hop summary prompt* variant. This approach utilizes an iterative and recursive summarization process to compress information from distant hops, enabling its inclusion without exceeding the token limit. For example, to construct a 3-hop summary prompt, the LLM first summarizes the immediate neighborhood of each 2-hop node (i.e., the 3-hop nodes). Next, it summarizes the neighborhood neighborhood of each 1-hop node, where 2-hop nodes are now represented by their previously generated summaries. The final prompt for the target query node thus includes the immediate neighborhood – containing these rich, summarized 1-hop neighbors, which carry compressed representations of the 2-hop and 3-hop topology and features. This allows the model to access 3-hop information that we could not have included before due to token limits.

We evaluate 2-hop, 3-hop, and 4-hop summary prompts on the long-text datasets in Table 8. Execution logs confirm that our ReAct-based modes do not query beyond 4 hops on these specific datasets, ensuring that evaluating up to 4-hop summaries provides a fair comparison of maximum resource exposure.

**Finding 13.** *Iterative neighborhood summarization hurts performance beyond a certain number of hops.*

The results in Table 8 yield several key insights. First, the 2-hop summary prompt consistently outperforms both the 1-hop prompt and the 2-hop budget prompt, establishing it as a much stronger and more token-efficient prompting baseline.

Second, we observe a clear trend of diminishing, and even negative, returns as we extend this summarization to more distant hops. Performance peaks at the 2-hop summary and then slightly declines with the 3-hop summary and 4-hop summary (Finding 13). Since we know adding hops is beneficial in these homophilic datasets, this decline indicates that the iterative summarization process is lossy. Each summary of a summary compounds the information loss, eventually outweighing the benefit of including more distant nodes.

Table 8: Accuracy of LLM-graph interaction modes Prompting, Summary Prompting, GraphTool, and Graph-as-Code on long-text datasets. Best per-dataset results are **bold**, runner-up underlined.

|  | citeseer | reddit | computer | photo | instagram | wiki-cs |
|---|---|---|---|---|---|---|
| 0-hop prompt | 68.20 | 47.90 | 65.50 | 69.80 | 48.00 | 74.00 |
| 1-hop prompt | 68.30 | 59.30 | 86.10 | 85.80 | 56.10 | TokenLimit |
| 2-hop prompt | 69.40 | TokenLimit | TokenLimit | TokenLimit | TokenLimit | TokenLimit |
| 2-hop budget prompt | 70.20 | 54.40 | 86.00 | 85.60 | 54.50 | 80.80 |
| 2-hop summary prompt | 70.40 | 57.60 | **86.20** | 86.00 | 55.70 | 81.70 |
| 3-hop summary prompt | 69.80 | 57.00 | 85.80 | 85.60 | 55.80 | 81.00 |
| 4-hop summary prompt | 69.60 | 55.90 | 86.00 | 85.40 | 55.60 | 80.90 |
| GraphTool | 68.30 | 56.25 | 80.80 | 77.00 | 47.80 | 76.27 |
| Graph-as-Code | **71.80** | **61.60** | **86.20** | **86.40** | **56.40** | **82.20** |

**Finding 14.** *Graph-as-Code outperforms multi-hop summarization prompting by adaptively retrieving only the necessary information.*

Finally, even these stronger, multi-hop summary baselines are still outperformed by Graph-as-Code. This reinforces Finding 4, confirming that the superiority of Graph-as-Code is not merely a byproduct of "seeing more" information. Instead, Graph-as-Code adaptively and surgically retrieves the optimal information required for reasoning (Finding 14). For instance, the programmatic execution allows the model to retrieve only the labels of 2-hop neighbors (which are token-cheap) while ignoring their less informative features, or vice-versa. Prompting, even with iterative summarization, remains a non-adaptive, pre-computed information dump. We thus conclude that Graph-as-Code's programmatic, iterative approach allows it to execute a more flexible and efficient reasoning strategy, leading to its superior performance.

## D.2 GRAPH-AS-CODE'S RELIANCE ON STRUCTURE VS. CODE-GENERATION PRIORS

We verify that the strong performance of Graph-as-Code ($\phi_{\text{code}}$) is not merely an artifact of code-generation priors with a control experiment using randomly shuffled adjacency matrices. This process preserves all original node features and the graph's degree distribution, but destroys all meaningful structural patterns. We repeat our evaluations across 7 representative datasets and report the average accuracy over 10 independent random adjacency permutations.

Table 9: Accuracy of baselines and LLM-graph interaction modes Prompting, GraphTool and Graph-as-Code on datasets with randomly shuffled adjacency matrices.

|  | cora | arxiv | cornell | texas | citeseer | photo | wiki-cs |
|---|---|---|---|---|---|---|---|
| Random | 14.13 | 2.44 | 21.74 | 8.40 | 16.80 | 8.20 | 9.90 |
| Label propagation | 13.29 | 3.85 | 30.48 | 10.34 | 13.10 | 8.40 | 9.60 |
| 0-hop prompt | 64.21 | 68.10 | 81.57 | 53.20 | 68.20 | 69.80 | 74.00 |
| 1-hop prompt | 58.10 | 61.50 | 74.30 | 47.10 | 60.30 | 63.20 | TokenLimit |
| 2-hop prompt | 44.50 | 47.20 | 60.10 | TokenLimit | 45.10 | TokenLimit | TokenLimit |
| GraphTool | 56.20 | 59.80 | 72.40 | 45.90 | 58.10 | 61.10 | 66.20 |
| Graph-as-Code | 56.50 | 59.10 | 72.90 | 45.20 | 58.80 | 60.90 | 66.70 |

As shown in Table 9, the performance of all structure-aware methods significantly decreases when compared to their original results in Tables 1 to 3. This confirms that they are attempting to use the neighborhood information, which now only adds noise. These additional trends arise:

1. The 0-hop prompt's performance is unaffected, since it never queries the neighborhood. It is thus immune to structural noise and becomes the best-performing baseline.

2. Label Propagation is on par with a random guess. This is expected, as it relies exclusively on the now-meaningless structural information.

3. Performance degradation scales with the amount of noise introduced. 2-hop prompting is impacted more severely compared to 1-hop prompting, as it ingests the largest volume of noisy neighborhood context.

4. GraphTool and Graph-as-Code prove more robust than the naive 2-hop prompt. This suggests that while the LLM's reasoning is confused by the noise, it can still mitigate the noise better than a simple prompt expansion, likely by weighing its own (correct) features more heavily than its neighbors' (noisy) information.

**These results strongly indicate that the superior performance of Graph-as-Code and other interaction modes seen in our main paper is not an artifact of code-generation priors**, but a direct result of effectively utilizing the true graph structure. This experiment also reinforces Finding 6. While 100% random noise hurts all methods, Graph-as-Code's performance degrades less severely than the 2-hop prompt, demonstrating a superior ability to handle noisy or irrelevant structural information, even in this extreme scenario.

### D.3  ALGORITHMIC REASONING EVALUATION

We test algorithmic reasoning capabilities, with an additional experiment on a synthetic shortest path prediction task.

**Synthetic dataset generation.** We generated a dataset of 100 synthetic Erdős-Rényi graphs, each comprising 50 nodes, with the edge probability chosen to ensure full connectivity. For each graph, we randomly sampled 100 source-target node pairs. The regression task is to predict the integer length of the shortest path ($k$) between the two nodes. This setup allows us to assess predictive performance as a direct function of problem complexity, represented here by the true path length $k$. We report the Mean Squared Error (MSE) for 2-hop Prompting, GraphTool+, and Graph-as-Code across varying path lengths in Table 10.

Table 10: Mean Squared Error (MSE) of LLM-graph interaction modes Prompting, GraphTool and Graph-as-Code on the synthetic shortest path prediction task across varying true path lengths ($k$).

| Method | $k = 1$ | $k = 2$ | $k = 3$ | $k = 4$ | $k \geq 5$ | $k \geq 1$ |
|---|---|---|---|---|---|---|
| 2-hop prompt | 0.00 | 0.00 | 2.02 | 2.29 | 2.05 | 2.82 |
| GraphTool+ | 0.00 | 0.00 | 0.74 | 2.46 | 2.37 | 2.13 |
| Graph-as-Code | 0.00 | 0.00 | 0.00 | 0.00 | 0.00 | 0.00 |

As shown in Table 10, the 2-hop Prompting method achieves a perfect prediction for path lengths $k = 1$ and $k = 2$. This is expected, as the target node naturally falls within the 2-hop neighborhood provided in the static prompt, allowing the LLM to simply "find" the answer in its context. However, for any path length $k \geq 3$, the target lies outside the prompt's fixed context boundary. Lacking any mechanism to explore further, the LLM is forced to guess, and its MSE increases dramatically. This confirms that static prompting is incapable of the iterative computation needed for path-finding algorithms.

Similarly, GraphTool+ performs perfectly for path lengths of $k = 1$ and $k = 2$. However, its performance begins to degrade at $k = 3$ and becomes highly error-prone for longer paths, with the MSE jumping to 2.46 at $k = 4$. This indicates that while GraphTool+ can attempt to simulate a Breadth-First Search (BFS) by iteratively traversing the graph via tool calls, the computational space of possible paths explodes as length increases, making the LLM highly error-prone as expected.

Finally, Graph-as-Code ($\phi_{\text{code}}$) achieves perfect prediction across all evaluated path lengths. This flawless performance occurs for a simple reason: the LLM does not need to internally simulate the path-finding algorithm step-by-step. Instead, it leverages its code-generation capabilities to write standard BFS execution code, which the environment runs deterministically. This experiment demonstrates that **for pure algorithmic reasoning, delegating computation to generated code as in Graph-as-Code is vastly superior to other interaction modes**.

D.4 COMPARISON WITH SUPERVISED ARCHITECTURES

While the primary focus of our paper is evaluating the capabilities of LLMs at inference time, it remains meaningful to contextualize their performance against state-of-the-art supervised architectures. To this end, we extend our evaluation on long-text datasets to compare our zero-shot interaction modes against end-to-end GNNs – GCN (Kipf & Welling, 2017b), GAT (Veličković et al., 2018) and hybrid LLM+GNN models – Encoder GCN (Wu et al., 2025), TAPE (He et al., 2024), LLaGA (Chen et al., 2024), and UniGraph (He et al., 2025). All GNN and hybrid LLM+GNN results were taken from Wu et al. (2025).

Table 11: Accuracy comparison between classic GNNs, hybrid LLM+GNNs, and our evaluated LLM-graph interaction modes Prompting, GraphTool and Graph-as-Code on long-text datasets.

| Method Type | Model | citeseer | reddit | computer | photo | instagram |
|---|---|---|---|---|---|---|
| Classic GNN | GCN | 70.55 | 61.44 | 71.44 | 69.25 | 63.50 |
| | GAT | 69.94 | 60.60 | 83.39 | 80.40 | 63.56 |
| Hybrid LLM+GNN | Encoder GCN | 71.39 | 68.65 | 88.22 | 84.84 | 67.49 |
| | TAPE | 71.87 | 62.43 | 89.52 | 86.46 | 66.07 |
| | LLaGA | 74.15 | 69.20 | 88.80 | 87.10 | 68.20 |
| | UniGraph | 72.50 | 68.80 | 88.30 | 85.90 | 67.60 |
| LLM-Only | 0-hop prompt | 68.20 | 47.90 | 65.50 | 69.80 | 48.00 |
| | 1-hop prompt | 68.30 | 59.30 | 86.10 | 85.80 | 56.10 |
| | 2-hop budget prompt | 70.20 | 54.40 | 86.00 | 85.60 | 54.50 |
| | GraphTool+ | 68.70 | 61.80 | 83.10 | 81.30 | 48.20 |
| | Graph-as-Code | 71.80 | 61.60 | 86.20 | 86.40 | 56.40 |

This comparison shows that hybrid LLM+GNN models generally outperform both classic GNNs and LLM-only methods. We hypothesize this is because they effectively combine the distinct strengths of each architecture: Classic GNNs are specialized for structural processing and operate on shallow, low-dimensional text features (e.g., bag-of-words, TF-IDF); LLM-Only methods excel at deeply understanding raw, rich text but are not specialized for graph-based message passing.

E AN EXAMPLE OF GRAPH-AS-CODE REASONING

To demonstrate how the Graph-as-Code ($\phi_{\text{code}}$) mode operates in practice, we present a representative execution log in Example 1. The log illustrates the practical, step-by-step reasoning process of the Graph-as-Code mode. It shows the agent forming a hypothesis from its own features (Class 5, 'Reinforcement Learning') and then programmatically testing that hypothesis against its neighborhood – even noting the correlation between Class 1 and Class 5. This demonstrates a more complex and interactive reasoning process than in standard static prompting.

---

**Example 1 – Graph-as-Code execution log for node 133 of the cora dataset**

**Task:** You are solving a node-based reasoning task for node 133. You have a pandas DataFrame `df` where each row corresponds to a node, indexed by its `node_id`.

**Instructions:** Always begin with reasoning. After your reasoning, provide a single, valid pandas command on a new line.

**Schema structure:**
– The DataFrame index is the node id: `df.loc[node_id]`.
– Textual description: `df.loc[node_id, 'features']`.
– List of neighbor node IDs: `df.loc[node_id, 'neighbors']`.
– Integer node label (if in training set, otherwise `None`): `df.loc[node_id, 'label']`.

**Available class labels:**
```
0:  Rule Learning,
1:  Neural Networks,
2:  Case Based,
3:  Genetic Algorithms,
4:  Theory,
5:  Reinforcement Learning,
6:  Probabilistic Methods
```

---

**Request 1:** I need to classify node 133. I will start by inspecting its features and its label.
```
df.loc[133][['features', 'label']]
```

**Response 1:**
```
features:  'A Reinforcement Learning Approach to Job-Shop Scheduling'
label:  None
```

**Request 2:** The label is unknown. The features strongly suggest 'Reinforcement Learning' (class 5). I will check the labels of its neighbors to confirm this.
```
df.loc[df.loc[133, 'neighbors'], 'label']
```

**Response 2:**
```
node_id    label
134        5.0
135        5.0
707        5.0
1588       5.0
1589       NaN
2048       1.0
```

**Request 3:** I have 5 neighbors with known labels. Four are class 5 ('Reinforcement Learning') and one is class 1 ('Neural Networks'). The majority class among neighbors is 5. The presence of class 1 ('Neural Networks') also supports this, as reinforcement learning is highly correlated with neural networks. This neighborhood evidence strongly matches my initial hypothesis from the node's own features. I am confident in this prediction.
```
Answer:  5
```

---

# F    DETAILS ON LLM-GRAPH INTERACTION MODES

In this section, we provide the full prompt templates used for each LLM-graph interaction mode evaluated in our experiments. These templates specify the instructions, available actions, and response formats provided to the LLM during node classification tasks. Specifically, we include the exact text used for the Prompting, GraphTool+, and Graph-as-Code modes, in Templates 1 to 3, respectively, enabling reproducibility and facilitating future methodological comparisons.

---

**Template 1 – Prompting**

**Task:** You are solving a node-based task. Your goal is to determine the label for node `{node_id}`.
The final answer must be submitted as an integer corresponding to a class label. Below is the mapping from each integer index to its associated label.
**Available class labels:**
```
0:  {text description of label 0}
1:  {text description of label 1}
...
```
Node `{node_id}` has the textual description `{feat_id}` and belongs to label class `{label_id}` (or `None`).
Node `{node_id}` has the following neighbors 1-hop away:
  Node `{n_1}` has the textual description `{feat_n1}` and belongs to label class `{label_n1}` (or `None`).
  Node `{n_2}` has the textual description `{feat_n2}` and belongs to label class `{label_n2}` (or `None`).
  ...
Node `{node_id}` has the following neighbors 2-hops away:
  Node `{m_1}` has the textual description `{feat_m1}` and belongs to label class `{label_m1}` (or `None`).
  ...
Think and end your response with: `Answer:  [class_id]`.

---

**Template 2 – GraphTool+**

**Task:** You are solving a node-based reasoning task using interleaved steps. Your goal is to determine the label for node `{node_id}`. At each step, you may choose one of several available actions to gather information or submit your final prediction.
**Instructions:** Always begin with reasoning. You may take as many steps as needed, but aim to solve the task efficiently using the fewest necessary actions. Before each action, assess what information is available, what's missing, which action is most appropriate next, and how many steps likely remain. Then, on a new line, specify your chosen action using one of the formats below. It must be the final non-empty line of your response.
**Available actions:**
– `Action 0, answer class_id`: Submit your final answer as an integer label.
– `Action 1, node node_id`: Retrieve the list of neighboring nodes connected to the specified node.
– `Action 2, node node_id`: Retrieve the textual description (features) of the specified node.
– `Action 3, node node_id`: Retrieve the label of the specified node if it is in the training set; otherwise, return `None`.
– `Action 4, node node_id, hop num_hop`: Retrieve the textual descriptions (features) of all nodes that are exactly `num_hop` hops away from the specified node.
– `Action 5, node node_id, hop num_hop`: Retrieve the labels (or `None`) of all nodes that are exactly `num_hop` hops away from the specified node.
**Available class labels:**
```
0:  {text description of label 0}
...
```
Now begin your reasoning in the Scratchpad below:

Table 12: Statistics of all datasets

|  | Dataset | #Nodes | #Edges | #Classes | Train/Val/Test (%) |
|---|---|---|---|---|---|
| Short-text homophilic | cora | 2,708 | 5,429 | 7 | 60/20/20 |
|  | pubmed | 19,717 | 44,338 | 3 | 60/20/20 |
|  | arxiv | 169,343 | 1,166,243 | 40 | 53.7/17.6/28.7 |
|  | products | 2,449,029 | 61,859,140 | 47 | 8.0/1.6/90.4 |
| Heterophilic | cornell | 191 | 292 | 5 | 60/20/20 |
|  | texas | 187 | 310 | 5 | 60/20/20 |
|  | washington | 229 | 394 | 5 | 60/20/20 |
|  | wisconsin | 265 | 510 | 5 | 60/20/20 |
| Long-text homophilic | citeseer | 3,186 | 4,277 | 6 | 60/20/20 |
|  | reddit | 33,434 | 198,448 | 2 | 60/20/20 |
|  | computer | 87,229 | 721,081 | 10 | 60/20/20 |
|  | photo | 48,362 | 500,928 | 12 | 60/20/20 |
|  | instagram | 11,339 | 144,010 | 2 | 60/20/20 |
|  | wiki-cs | 11,701 | 215,863 | 10 | 60/20/20 |

---

**Template 3 – Graph-as-Code**

**Task:** You are solving a node-based reasoning... for node `{node_id}`. You have a pandas DataFrame `df` where each row corresponds to a node, indexed by its `node_id`.
**Instructions:** Always begin with reasoning...
**Schema structure:**
 – The DataFrame index is the node id. Access a row by node id with: `df.loc[node_id]`.
 – The column `features` stores each node's textual description: `df.loc[node_id, 'features']`.
 – The column `neighbors` stores a list of neighbor node IDs: `df.loc[node_id, 'neighbors']`.
 – The column `label` contains the integer node label if it belongs to the training set; otherwise `None`.
You may query ANY column(s) of `df` using any valid pandas command that applies to a DataFrame named `df`. You may also use `pd.*` utilities with `df` as input. The dataframe can be long, so you may want to avoid commands that print the entire table.
**Response format:**
 – For intermediate steps: reason then on the final line output a *single* valid pandas expression.
 – To finish: reason then on the final line respond exactly as: `Answer [class_id]`.
**Available class labels:**
```
0:  {text description of label 0}
...
```

All experiments were conducted using 8 Intel Xeon Platinum 8370C cpus.

## G DATASET STATISTICS

The statistics of all datasets can be found in Table 12.

