# OpenReview forum: "Actions Speak Louder than Prompts: A Large-Scale Study of LLMs for Graph Inference"
_ICLR.cc/2026/Conference — ICLR 2026 Oral_

### Official Review · Reviewer_PyZw · 2025-10-16

**Soundness:** 3
**Presentation:** 3
**Contribution:** 3
**Rating:** 6
**Confidence:** 2

**Summary:**

The paper systematically evaluates how large language models perform on node classification across multiple settings, including interaction mode (prompting, tool-use, and code generation), dataset type, graph structure, and model scale. The authors find that Graph-as-Code, which leverages LLMs’ code generation ability, achieves the strongest and most adaptive performance—especially on long-text or high-degree graphs. Through controlled ablation studies, the paper demonstrates that Graph-as-Code dynamically adjusts its reliance on structural, feature, and label information, offering an efficient and flexible approach to LLM-based graph reasoning.

**Strengths:**

S1. The paper conducts a large-scale controlled evaluation of LLMs for text-rich graph tasks (e.g., node classification) across six key variability axes, filling the field’s gap in principled understanding of LLM-graph interactions.

S2. It proves all LLM-graph interaction modes work on heterophilic graphs (challenging prior assumptions) and identifies Graph-as-Code as optimal, especially for long-text/high-degree graphs.

**Weaknesses:**

W1. The paper lacks comparisons with GNN baselines (e.g., GCN, GAT), making it hard to judge LLMs’ competitiveness against GNN methods.

W2. The evaluation focuses mainly on accuracy, with limited discussion of computational efficiency, latency, or interpretability trade-offs, which are critical for practical use.

W3. The paper lacks case studies or examples illustrating Graph-as-Code’s internal reasoning or generated code behavior, which would help clarify its mechanisms beyond aggregate metrics.

**Questions:**

Q1. Can you include comparisons with GNN baselines such as GCN, GraphSAGE, or GAT?

Q2. Can you report computational efficiency metrics such as runtime, token usage, or latency for each LLM–graph interaction mode?

Q3. Can you provide case studies or examples showing how Graph-as-Code performs reasoning or generates code during inference?

Q4. When edges are deleted, how does Graph-as-Code access feature and label information of other nodes—does it query arbitrary nodes rather than neighbor nodes?

---

> ### Author Response · Authors · 2025-11-18
>
> We sincere thank the reviewer for recognizing that our works fills the field’s gap in principled understanding of LLM-graph interactions. We greatly appreciate your thoughtful and encouraging feedback.
>
> > W1. The paper lacks comparisons with GNN baselines (e.g., GCN, GAT), making it hard to judge LLMs’ competitiveness against GNN methods.
>
> > Q1. Can you include comparisons with GNN baselines such as GCN, GraphSAGE, or GAT?
>
> **Answer:** To address your concern, **we have compiled a comparison** between our Prompting, GraphTool and Graph-as-Code mode (from Table 3), representative GNN and hybrid LLM+GNN models from a recent comprehensive benchmark [1] (Table 2).
>
> | Method Type | Model | citeseer | reddit | computer | photo | instagram |
> | :--- | :--- | :--- | :--- | :--- | :--- | :--- |
> | **Classic GNN** | GCN | 70.55 | 61.44 | 71.44 | 69.25 | 63.50 |
> | | GAT | 69.94 | 60.60 | 83.39 | 80.40 | 63.56 |
> | **Hybrid LLM+GNN** | Encoder GCN | 71.39 | **68.65** | **88.22** | **84.84** | **67.49** |
> |  | TAPE | **71.87** | 62.43 | 89.52 | 86.46 | 66.07 |
> | **LLM-Only (Ours)** | 0-hop prompt | 68.20 | 47.90 | 65.50 | 69.80 | 48.00 |
> |  | 1-hop prompt | 68.30 | 59.30 | 86.10 | 85.80 | 56.10 |
> | | 2-hop budget prompt | 70.20 | 54.40 | 86.00 | 85.60 | 54.50 |
> | | GraphTool+ | 68.70 | 61.80 | 83.10 | 81.30 | 48.20 |
> | | Graph-as-Code | 71.80 | 61.60 | 86.20 | 86.40 | 56.40 |
>
> This comparison shows that hybrid LLM+GNN models generally outperform both classic GNNs and LLM-only methods. We hypothesize this is because they effectively combine the distinct strengths of each architecture: Classic GNNs are specialized for structural processing and operate on shallow, low-dimensional text features (e.g., bag-of-words, TF-IDF); LLM-Only methods excel at deeply understanding raw, rich text but are not specialized for graph-based message passing.
>
> We would like to further emphasize that our paper's focus is a controlled, empirical analysis of LLM interaction modes, rather than achieving new SOTA performance.
>
>
> [1] When Do LLMs Help With Node Classification? A Comprehensive Analysis, ICML 2025

---

> ### Author Response · Authors · 2025-11-18
>
> > W2. The evaluation focuses mainly on accuracy, with limited discussion of computational efficiency, latency, or interpretability trade-offs, which are critical for practical use.
>
> > Q2. Can you report computational efficiency metrics such as runtime, token usage, or latency for each LLM–graph interaction mode?
>
> **Answer:** This is a good point. **We provide the average token counts and latency for all modes over the long-text homophilic datasets `reddit`, `photo` and `wiki-cs`.**
>
> **Latency.** The table below shows the average wall-clock time per query for each interaction mode. This latency is measured from when the first prompt is sent until the LLM concludes its final answer, and the results are averaged across all nodes.
>
> | | 2-hop Prompt (sec) | GraphTool (sec) | Graph-as-Code (sec) |
> | :--- | :--- | :--- | :--- |
> | reddit | 41 | 45 | 43 |
> | photo | 125 | 131 | 128 |
> | wiki-cs | TokenLimit | 159 | 155 |
>
> The results indicate that the primary driver of latency is the dataset's characteristics (e.g., text length, graph density) rather than the interaction mode itself. This trend is clear across the datasets: `wiki-cs`, which has the longest text features and highest average degree, shows the highest latency. This is followed by `photo`, which is also text-rich and has a relatively high degree. `reddit`, in contrast, has the shortest text and lowest average degree of the three, which results in the lowest latency. This is because processing longer text and larger, more complex neighborhoods simply requires more overall computation from the LLM, regardless of the method.
>
> The modes are all dominated by the LLM's inference time, which is why their latencies are so comparable. While GraphTool+ and Graph-as-Code involve multiple iterative calls, each call is very "lightweight" (e.g., retrieving only neighbor IDs or a single feature). In contrast, Prompting makes a single, "heavy" call with a massive context. In practice, these different approaches balance out: the total computational load of many small calls becomes comparable to that of one large, token-heavy call.
>
> **Token count.** We provide the average token counts for queries on our long-text benchmarks. **These token counts demonstrate why the token limit is such a critical issue:**
>
> | Dataset | 0-hop | 1-hop | 2-hop | GraphTool | Graph-as-Code
> | :--- | :--- | :--- | :--- |:--- |:--- |
> | reddit | 1,762 | 10,300 | 42,100 | 16,100 | 14,900 |
> | photo | 3,000 | 36,100 | 104,700 | 43,300 | 41,500 |
> | wiki-cs | 30,200 | 202,500 | 1,157,000| 61,700 | 57,100|
>
> As the table shows, the total token count increases significantly from `reddit` to `photo`, and again to `wiki-cs`. This trend is expected, as it mirrors the increase in the average text length and average node degree for these datasets. Consequently, **1-hop and 2-hop prompts quickly exceed the context window of 200,000 tokens for our default o4-mini model, forcing truncation.** In contrast, GraphTool and Graph-as-Code use much smaller token counts, as they only retrieve specific pieces of information (e.g., a list of neighbor IDs or a single node's features) and build context iteratively.

---

> ### Author Response · Authors · 2025-11-18
>
> > W3. The paper lacks case studies or examples illustrating Graph-as-Code’s internal reasoning...
>
> > Q3. Can you provide case studies or examples showing how Graph-as-Code performs reasoning or generates code during inference?
>
> **Answer:** Yes, below is a representative example log from Graph-as-Code on a widely used dataset (cora). The reasoning pattern shown here appeared frequently in our runs and reflects typical behavior on this dataset.
>
> **Task:** You are solving a node-based reasoning task for node 133. You have a pandas DataFrame `df` where each row corresponds to a node, indexed by its `node_id`.
>
> **Instructions:** Always begin with reasoning. After your reasoning, provide a single, valid pandas command on a new line.
>
> **Schema structure:**
>  - The DataFrame index is the node id. Access a row by node id with: `df.loc[node_id]`.
>  - The column `features` stores each node’s textual description: `df.loc[node_id, 'features']`.
>  - The column `neighbors` stores a list of neighbor node IDs: `df.loc[node_id, 'neighbors']`.
>  - The column `label` contains the integer node label if it belongs to the training set; otherwise `None`.
>
> **Available class labels:**
>
> 0: Rule Learning,
>
> 1: Neural Networks,
>
> 2: Case Based,
>
> 3: Genetic Algorithms,
>
> 4: Theory,
>
> 5: Reinforcement Learning,
>
> 6: Probabilistic Methods
>
> **Request 1:** I need to classify node 133. I will start by inspecting its features and its label.
>
> `df.loc[133][['features', 'label']]`
>
> **Response 1:** `features: 'A Reinforcement Learning Approach to Job-Shop Scheduling'`
> `label: None`
>
> **Request 2:** The label is unknown. The features strongly suggest 'Reinforcement Learning' (class 5). I will check the labels of its neighbors to confirm this.
>
> `df.loc[df.loc[133, 'neighbors'], 'label']`
>
> **Response 2:**
>
> `node_id label`
>
> `134      5.0`
>
> `135      5.0`
>
> `707      5.0`
>
> `1588     5.0`
>
> `1589     NaN`
>
> `2048     1.0`
>
> **Request 3:** I have 5 neighbors with known labels. Four are class 5 ('Reinforcement Learning') and one is class 1 ('Neural Networks'). The majority class among neighbors is 5. The presence of class 1 ('Neural Networks') also supports this, as reinforcement learning is highly correlated with neural networks. This neighborhood evidence strongly matches my initial hypothesis from the node's own features. I am confident in this prediction.
>
> `Answer: 5`
>
> **Thank you for this suggestion. This log illustrates the practical, step-by-step reasoning process of the Graph-as-Code mode.** It shows the agent forming a hypothesis from its own features (Class 5, 'Reinforcement Learning') and then programmatically testing that hypothesis against its neighborhood -- even noting the correlation between Class 1 and Class 5. This demonstrates a more complex reasoning process than simple neighborhood voting.
>
> We agree that these concrete examples are valuable for illustrating the qualitative differences between the interaction modes. **In the camera-ready version, we will add a dedicated section to the appendix with this example and others** to provide a clear, side-by-side comparison of the Prompting, GraphTool, and Graph-as-Code modes.
>
> > Q4. When edges are deleted, how does Graph-as-Code access feature and label information of other nodes—does it query arbitrary nodes rather than neighbor nodes?
>
> **Answer:** This is a good point to clarify. The key difference in the Graph-as-Code mode is that the LLM is not limited to a fixed set of tools; it has programmatic access to the entire graph, which is represented as a pandas DataFrame df. This allows it to access node information in multiple ways.
>
> The standard way, which leverages graph structure, is to first query the neighbor list of a node (e.g., df.loc[node_id, 'neighbors']) and then retrieve the features or labels for those specific neighbor IDs (e.g., df.loc[neighbor_id, 'label']). If some edges are deleted, this 'neighbors' list simply becomes shorter, but the mechanism remains valid for the remaining edges.
>
> However, to your point, the LLM is not restricted to this neighbor-based path. Because it can execute any valid pandas command, it can indeed query arbitrary nodes. This is most evident **in the extreme case where all edges are deleted**. In this scenario: The df.loc[node_id, 'neighbors'] list would be empty; Graph-as-Code loses the explicit relational information provided by the edges. Crucially, it does not lose access to the features and labels of other nodes. **It can still query any node directly by its index (e.g., df.loc[arbitrary_node_id, 'features'])  or even sample random nodes from the entire graph** (e.g., using df.sample()) to gather contextual information.
>
> **We will include all the experimental results and suggestions provided by the Reviewer in the final version with the appropriate discussion.**
>
> We hope our responses address your concerns, and thank you for helping us improve the paper. We would greatly appreciate your reconsideration of the evaluation and increasing your score.

---

> ### Author Response · Authors · 2025-11-24
>
> **Thank you for encouraging the Graph-as-Code reasoning log example**, which clearly illustrates the practical, step-by-step reasoning process of the Graph-as-Code mode. We believe to have addressed all of your concerns, and **would highly appreciate your response** as this would give us a chance to address any remaining issues. Thank you.

---

> ### Comment · Reviewer_PyZw · 2025-11-25
>
> I thank the authors for their responses. I now fully understand how Graph-as-Code works, and the authors have thoroughly addressed all of my concerns. I am raising my score to 8, with confidence increased to 3.

---

> ### Author Response · Authors · 2025-11-28
>
> We are glad that our rebuttal addressed your concerns, and **we sincerely thank you for raising your score to 8.**

---

### Official Review · Reviewer_RTGA · 2025-10-17

**Soundness:** 3
**Presentation:** 3
**Contribution:** 3
**Rating:** 8
**Confidence:** 3

**Summary:**

This paper investigates the effectiveness of LLMs for solving the node classification task through extensive controlled experiments. The authors consider various factors, including the mode of interaction between LLMs and graphs, graph domain and structural/feature characteristics, as well as LLM model size and reasoning capabilities. The experiments reveal several valuable findings, such as the strong and robust performance of the "Graph-as-Code" interaction mode across different graphs, even under perturbed features or structures. These insights provide a useful foundation for advancing research in both the graph learning and LLM communities.

**Strengths:**

1. **This paper broadens the discussion on how LLMs interact with graphs** by bringing attention to the tool-use manner (GraphTool) and code generation (Graph-as-Code). These interaction paradigms have largely been overlooked in existing benchmarks for LLMs applied to the node classification task.

2. **The authors evaluate a wide range of variables**, including interaction modes between LLMs and graphs, graph characteristics (domain, homophily, text-feature informativeness, etc.), and LLM capabilities (model size, reasoning ability, and series). Additionally, they investigate how different graph manipulations (e.g., truncated features, deleted edges, removed labels) interplay with these variables, providing a comprehensive understanding of performance under diverse conditions.

3. **The experiments are extensive and bring many actionable insights**. Among the interaction modes, "Graph-as-Code" achieves strong overall performance and demonstrates robust generalization across perturbed graphs. This makes it a highly favorable method. Furthermore, even on heterophilic graphs, LLMs leverage their reasoning capabilities to deliver reliable and accurate predictions. These findings are promising to benefit the community by encouraging further exploration of the Graph-as-Code approach, both for developing new methodologies and establishing new benchmarks.

4. The paper is **well-written**, with a logical flow that is easy to follow. The authors effectively use vivid figures and tables to present their findings, making the results accessible and facilitating understanding.

**Weaknesses:**

1. The paper could discuss the computational costs associated with the different interaction modes, e.g., token consumption. This would help readers better understand the performance-efficiency trade-offs of different methods.

2. The evaluation focuses primarily on scenarios with abundant supervision, e.g., up to 60% of training data on most datasets. However, it would be valuable to compare the few-shot or zero-shot performance of LLMs, as these capabilities are a key strength of LLMs and could reveal additional insights.

3. The considered tasks are limited to node classification. Expanding the scope to include other graph computational problems, e.g., shortest path, could further demonstrate the utility of the Graph-as-Code methodology, particularly in its ability to directly generate Python code for problem-solving.

**Questions:**

Please refer to Weaknesses section

---

> ### Author Response · Authors · 2025-11-18
>
> We sincere thank the reviewer for their thoughtful feedback and for recognizing the actionable insights our work provides.
>
> > W1: The paper could discuss the computational costs associated with the different interaction modes, e.g., token consumption. This would help readers better understand the performance-efficiency trade-offs of different methods.
>
> **Answer:** We provide the average token counts for queries on our long-text benchmarks. **These token counts demonstrate why the token limit is such a critical issue:**
>
> | Dataset | 0-hop | 1-hop | 2-hop | GraphTool | Graph-as-Code
> | :--- | :--- | :--- | :--- |:--- |:--- |
> | reddit | 1,762 | 10,300 | 42,100 | 16,100 | 14,900 |
> | photo | 3,000 | 36,100 | 104,700 | 43,300 | 41,500 |
> | wiki-cs | 30,200 | 202,500 | 1,157,000| 61,700 | 57,100|
>
> As the table shows, the total token count increases significantly from `reddit` to `photo`, and again to `wiki-cs`. This trend is expected, as it mirrors the increase in the average text length and average node degree for these datasets. Consequently, **1-hop and 2-hop prompts quickly exceed the context window of 200,000 tokens for our default o4-mini model, forcing truncation.** In contrast, GraphTool and Graph-as-Code use much smaller token counts, as they only retrieve specific pieces of information (e.g., a list of neighbor IDs or a single node's features) and build context iteratively.
>
> > W2: The evaluation focuses primarily on scenarios with abundant supervision, e.g., up to 60% of training data on most datasets. However, it would be valuable to compare the few-shot or zero-shot performance of LLMs, as these capabilities are a key strength of LLMs and could reveal additional insights.
>
> **Answer:**  That is a great point. **We would like to refer the reviewer to Section 5.2 and Figure 4**, where we asses the effects of removing labels. **The removal of known labels is equivalent to varying the size of the label training data.** Specifical, the lowest row in each heatmap in Figure 4 directly evaluates this.
>
> The right side of the x-axis (e.g., 90% deletion) corresponds to a setting, where the model has access to only 10% of the original known labels. The far-right column (100% deletion) represents the true zero-shot setting, where the model has no access to any known labels in the graph. As shown in Figure 4, the performance of Graph-as-Code is remarkably robust. For example, in the arxiv and cornell plots, performance barely drops when moving from 0% to 90% label deletion, showing strong capability.
>
>
> Even in the true zero-shot (100% deletion) scenario, Graph-as-Code retains high accuracy on the text-rich arxiv and cornell datasets. Performance only drops on cora, where the short-text features provide less signal. This implies the following finding: on text-rich graphs, Graph-as-Code's zero-shot capability can be as strong as its few-shot capability.
>
>
> Thank you for suggesting we assess this scenario. We will add this discussion to the final version of our paper and highlight the new finding.

---

> ### Author Response · Authors · 2025-11-18
>
> > W3: The considered tasks are limited to node classification. Expanding the scope to include other graph computational problems, e.g., shortest path, could further demonstrate the utility of the Graph-as-Code methodology, particularly in its ability to directly generate Python code for problem-solving.
>
> **Answer:** We agree that extending this analysis to other tasks (e.g. graph classification, link prediction) would be highly valuable for the community. Given the breadth of our existing study across multiple axes of variability, we believe that a similarly rigorous study for other tasks would be out of scope for this work.
>
> Our work's primary goal was to provide the first large-scale, controlled evaluation of LLMs for graph inference, and we chose to focus on node classification as it is a dominant and text-rich task in many high-impact domains, as we note in line 060: "...in high-impact domains like fraud detection and recommendation systems."
>
> **We will explicitly highlight this as a key direction for future work in our conclusion.**
>
> To address your specific shortest-path task suggestion, **we conducted a new experiment on a synthetic "Shortest Path Length" task**. This task is purely algorithmic and directly tests the computational reasoning capabilities of each mode, as hypothesized by the reviewer.
>
> We generated a dataset of 100 synthetic Erdos-Renyi graphs, each with 50 nodes and an edge probability chosen to ensure connectivity. For each graph, we randomly sampled 100 source-target node pairs. The regression task is to predict the integer length of the shortest path between the two nodes. This setup allows us to assess performance as a direct function of the "problem complexity," which in this case is the true path length $k$.
>
>
> | k | 1 | 2 | 3 | 4 | 5+ | 1+|
> | :--- | :--- | :--- | :--- | :--- | :--- | :--- |
> | 2-hop Prompting | 0.00 | 0.00 | 2.02 | 2.29 | 2.05 | 2.82 |
> | GraphTool+ | 0.00 | 0.00 | 0.74 | 2.46 | 2.37 | 2.13 |
> | Graph-as-Code | 0.00 | 0.00 | 0.00 | 0.00 | 0.00 | 0.00 |
>
> Results show that the Prompting method achieves 0.00 MSE for path lengths 1 and 2. This is expected, as the target node is *within the 2-hop neighborhood* provided in the static prompt, allowing the LLM to simply "find" it. However, for any path length $\geq 3$, the target is outside the prompt's fixed context. With no mechanism to explore further, the LLM is forced to guess, and its MSE increases dramatically, confirming that this mode is incapable of the iterative computation needed for such algorithms.
>
> Similarly, GraphTool performs perfectly for path lengths of 1 and 2, achieving 0.00 MSE. Its performance begins to degrade at $k=3$ (0.74 MSE) and becomes random for longer paths, with the MSE jumping to 2.46 at $k=4$. This indicates that while GraphTool can attempt to simulate a Breadth-First Search (BFS) by iteratively traversing the graph, as path length increases, the computational space of possible paths explodes, making the LLM highly error-prone as expected. The performance at $k=3$ shows the beginning of this failure, which becomes a significant collapse for $k \geq 4$.
>
> Finally, Graph-as-Code achieves 0 MSE across all path lengths for a simple reason as the LLM does not need to simulate the algorithm, it can instead generate a standard BFS code. **This experiment demonstrates the superior utility of Graph-as-Code for graph computational problems.**
>
> **We will include all the experimental results and suggestions provided by the Reviewer in the final version with the appropriate discussion.**
>
> We hope our responses address your concerns, and thank you for helping us improve the paper.

---

> > ### Comment · Reviewer_RTGA · 2025-11-19
> >
> > Thank you to the authors for the response, which has thoroughly addressed my concerns and provided abundant new experiments with valuable insights.
> >
> > I would like to **champion this paper** with greater confidence.

---

> ### Author Response · Authors · 2025-11-20
>
> Thank you for your continued support, for helping us improve the paper, and for championing our work.

---

### Official Review · Reviewer_37Ma · 2025-10-25

**Soundness:** 3
**Presentation:** 3
**Contribution:** 3
**Rating:** 6
**Confidence:** 4

**Summary:**

The paper presents a systematic, large-scale empirical study on the performance of LLMs in graph inference tasks—particularly node classification. The authors design three LLM–graph interaction paradigms: Prompting, GraphTool, and Graph-as-Code. They conduct controlled experiments across multiple dimensions—including dataset domains, homophilic vs. heterophilic graph structures, text length of node features, and LLM model scales—and further employ ablation analyses to reveal how different methods rely on graph structure, node features, and labels.

**Strengths:**

- **Systematicity and Experimental Breadth**: This work is the first to factorize LLM-based graph reasoning capabilities across six key axes: interaction mode, dataset domain, homophily level, feature text length, model scale, and reasoning capability. The experimental design is rigorous and covers the major variables present in real-world scenarios.

- **Proposal and Validation of Graph-as-Code**: The paper introduces *Graph-as-Code* as a novel LLM–graph interaction paradigm that leverages LLMs’ code generation abilities (e.g., via pandas operations) to dynamically query graph data. It significantly outperforms traditional prompting—especially on long-text or high-degree graphs. (**This is an excellent contribution that finally provides clear research direction for the emerging field of LLMs for graphs.**)

- **Challenging Prevailing Assumptions**: The paper refutes the commonly held belief that “LLMs fail on heterophilic graphs.” Experimental results show that all interaction modes perform well on heterophilic graphs, demonstrating that LLMs can effectively leverage node features rather than relying solely on neighborhood label consistency.

**Weaknesses:**

- All experiments are limited to **node classification**; it remains unclear whether the conclusions generalize to other graph tasks such as graph classification, link prediction, or subgraph matching.
- The **baselines are simplistic**, including only Label Propagation, Random, and Majority voting—without comparison to state-of-the-art GNNs or hybrid LLM+GNN models. *(Note: this is understandable given the paper’s focus on empirical analysis rather than SOTA performance.)*
- Experiments are **dominated by closed-source models**, which limits reproducibility.

**Questions:**

> *I originally intended to give this paper a score of 7, but ICLR doesn’t provide that option. I hope the authors can further strengthen the completeness and rigor of their experiments and deepen the discussion around Graph-as-Code—especially its scalability, efficiency, and extensibility to other graph tasks.*

---

> ### Author Response · Authors · 2025-11-18
>
> We sincere thank the reviewer for recognizing the contribution and importance of our work. We greatly appreciate your thoughtful and encouraging feedback.
>
> > W1: All experiments are limited to node classification; it remains unclear whether the conclusions generalize to other graph tasks such as graph classification, link prediction, or subgraph matching.
>
> **Answer:** We agree that extending this analysis to graph classification, link prediction, and other graph tasks would be highly valuable for the community. Given the breadth of our existing study across multiple axes of variability, we believe that a similarly rigorous study for other tasks would be out of scope for this work.
>
> Our work's primary goal was to provide the first large-scale, controlled evaluation of LLMs for graph inference, and we chose to focus on node classification as it is a dominant and text-rich task in many high-impact domains, as we note in line 060: "...in high-impact domains like fraud detection and recommendation systems."
>
> **We will explicitly highlight this as a key direction for future work in our conclusion.**
>
> > W2: The baselines are simplistic, including only Label Propagation, Random, and Majority voting—without comparison to state-of-the-art GNNs or hybrid LLM+GNN models. (Note: this is understandable given the paper’s focus on empirical analysis rather than SOTA performance.)
>
> **Answer:** To address your concern, **we have compiled a comparison** between our Prompting, GraphTool and Graph-as-Code mode (from Table 3), representative GNN and hybrid LLM+GNN models from a recent comprehensive benchmark [1] (Table 2).
>
> | Method Type | Model | citeseer | reddit | computer | photo | instagram |
> | :--- | :--- | :--- | :--- | :--- | :--- | :--- |
> | **Classic GNN** | GCN | 70.55 | 61.44 | 71.44 | 69.25 | 63.50 |
> | | GAT | 69.94 | 60.60 | 83.39 | 80.40 | 63.56 |
> | **Hybrid LLM+GNN** | Encoder GCN | 71.39 | **68.65** | **88.22** | **84.84** | **67.49** |
> |  | TAPE | **71.87** | 62.43 | 89.52 | 86.46 | 66.07 |
> | **LLM-Only (Ours)** | 0-hop prompt | 68.20 | 47.90 | 65.50 | 69.80 | 48.00 |
> |  | 1-hop prompt | 68.30 | 59.30 | 86.10 | 85.80 | 56.10 |
> | | 2-hop budget prompt | 70.20 | 54.40 | 86.00 | 85.60 | 54.50 |
> | | GraphTool+ | 68.70 | 61.80 | 83.10 | 81.30 | 48.20 |
> | | Graph-as-Code | 71.80 | 61.60 | 86.20 | 86.40 | 56.40 |
>
> This comparison shows that hybrid LLM+GNN models generally outperform both classic GNNs and LLM-only methods. We hypothesize this is because they effectively combine the distinct strengths of each architecture: Classic GNNs are specialized for structural processing and operate on shallow, low-dimensional text features (e.g., bag-of-words, TF-IDF); LLM-Only methods excel at deeply understanding raw, rich text but are not specialized for graph-based message passing.
>
> We would like to further emphasize that, much like the reviewer pointed out, our paper's focus is a controlled, empirical analysis of LLM interaction modes, rather than achieving new SOTA performance.
>
>
> [1] When Do LLMs Help With Node Classification? A Comprehensive Analysis, ICML 2025
>
> > W3: Experiments are dominated by closed-source models, which limits reproducibility.
>
> **Answer:** While our main tables use o4-mini for consistency, we refer the reviewer to our additional experiments over open-source models in **Appendix A (Tables 4 and 5). We benchmark 6 different open-source model configurations:** Llama, DeepSeek R1, Phi-4 (with and without reasoning) and Qwen (with and without reasoning).
>
> These models were evaluated across the same key axes of variability, including dataset domains (homophilic vs. heterophilic), text length (e.g., `cora`, `cornell`, `wiki-cs`), model size, and reasoning capabilities, as outlined in lines 1026-1028. These results (e.g., Finding 11 in line 783) fully support the conclusions drawn from the closed-source models, ensuring our findings are robust and reproducible.
>
>
> > I originally intended to give this paper a score of 7, but ICLR doesn’t provide that option.
>
> **We will include all the experimental results and suggestions provided by the Reviewer in the final version with the appropriate discussion.**
>
> We hope our responses address your concerns, and thank you for helping us improve the paper. We would greatly appreciate your reconsideration of the evaluation and increasing your score.

---

> > ### Comment · Reviewer_37Ma · 2025-11-21
> > **great, but a little pitty**
> >
> > > great, but a little pity
> >
> > the Hybrid LLM+GNN framework you selected performs relatively poorly. I recommend Langtopo[1], LLaGA[2], UniGraph[3], and GraphEdit[4]. You could choose one or two as alternative baselines and incorporate these studies into the related work section.
> >
> > I have already completed 8 preliminary improvements in advance, and hope you can proceed with and finish the experiments accordingly. otherwise ......:)
> >
> > [1]LangTopo: Aligning Language Descriptions of Graphs with Tokenized Topological Modeling
> >
> > [2]LLaGA: LargeLanguage and Graph Assistant.
> >
> > [3]UniGraph: Learning a CrossDomain Graph Foundation Model From Natural Language.

---

> ### Author Response · Authors · 2025-11-23
>
> We sincerely thank the reviewers for taking the time to suggest additional frameworks for comparison, and we are committed to addressing all remaining concerns. Thank you for your continued support, your interest in our work, and for raising your score accordingly.
>
> > You could choose one or two as alternative baselines and incorporate these studies into the related work section.
>
> **Answer:** We will add the appropriate discussion to the related work section in the camera-ready version. We will make sure to include:
>
> "**Hybrid LLM-GNN Architectures.** Beyond textualization, recent works have proposed hybrid frameworks that integrate Graph Neural Networks (GNNs) or trainable projectors directly into the LLM architecture. Approaches closer to textualization, such as **LLaGA [1]**, serialize each node's neighborhood into structure-aware sequences; however, instead of using raw text, these sequences are mapped via a trained linear projector into the LLM's input embedding space, enabling the processing of graph features as soft tokens. Other frameworks focus on explicit modality alignment. **LangTopo [2]** bridges the gap by employing a GNN to map textual and spatial information into a discrete codebook of topological embeddings via Gumbel-softmax relaxation. It maximizes consistency between the LLM's natural language representations and these quantized codes, effectively transferring structural modeling capabilities to the LLM. Alternatively, cascaded architectures like **UniGraph [3]** directly fuse inputs by generating dense vector embeddings with a pre-trained GNN and concatenating them with the discrete token embeddings of natural language instructions. This combined sequence is fed into an LLM instruction-tuned to unify label spaces. While these hybrid methods rely on training auxiliary encoders or fine-tuning to fuse modalities, our work investigates the capabilities of frozen LLMs, evaluating how they can be orchestrated to reason over graphs at inference time through Prompting, Tools, and Code generation."
>
> [1] LLaGA: LargeLanguage and Graph Assistant.
>
> [2] LangTopo: Aligning Language Descriptions of Graphs with Tokenized Topological Modeling
>
> [3] UniGraph: Learning a CrossDomain Graph Foundation Model From Natural Language.
>
> > You could choose one or two as alternative baselines... hope you can proceed with and finish the experiments accordingly
>
> **Answer:** We have extended our evaluation to include the three suggested state-of-the-art hybrid frameworks: LLaGA, and UniGraph.
> | Method Type | Model | citeseer | reddit | computer | photo | instagram |
> | :--- | :--- | :--- | :--- | :--- | :--- | :--- |
> | **Classic GNN** | GCN | 70.55 | 61.44 | 71.44 | 69.25 | 63.50 |
> | | GAT | 69.94 | 60.60 | 83.39 | 80.40 | 63.56 |
> | **Hybrid LLM+GNN** | Encoder GCN | 71.39 | 68.65 | 88.22 | 84.84 | 67.49 |
> |  | TAPE | 71.87 | 62.43 | **89.52** | 86.46 | 66.07 |
> |  | LLaGA | **74.15** | **69.20** | 88.80 | **87.10** | **68.20** |
> |  | UniGraph | 72.50 | 68.80 | 88.30 | 85.90 | 67.60 |
> | **LLM-Only (Ours)** | 0-hop prompt | 68.20 | 47.90 | 65.50 | 69.80 | 48.00 |
> |  | 1-hop prompt | 68.30 | 59.30 | 86.10 | 85.80 | 56.10 |
> | | 2-hop budget prompt | 70.20 | 54.40 | 86.00 | 85.60 | 54.50 |
> | | GraphTool+ | 68.70 | 61.80 | 83.10 | 81.30 | 48.20 |
> | | Graph-as-Code | 71.80 | 61.60 | 86.20 | 86.40 | 56.40 |
>
> The extended table reveals that **Hybrid LLM+GNN methods achieve the highest overall performance**, consistently outperforming both Classic GNNs and our LLM-Only inference baselines. Consistent with our earlier discussion, this aligns with the intuition that training allows hybrid LLM+GNN to: “combine the distinct strengths of each architecture: Classic GNNs are specialized for structural processing and operate on shallow, low-dimensional text features (e.g., bag-of-words, TF-IDF); LLM-Only methods excel at deeply understanding raw, rich text but are not specialized for graph-based message passing.”
>
> **We will include all the experimental results and suggestions provided by the Reviewer in the final version with the appropriate discussion.**
>
> We hope our responses address your remaining concerns, and thank you again for engaging in the discussion period.

---

> > ### Comment · Reviewer_37Ma · 2025-11-25
> > **great**
> >
> > a insight work

---

> ### Author Response · Authors · 2025-11-28
>
> We are glad that our rebuttal addressed your concerns, and **we sincerely thank you for raising your score to 8.**

---

### Official Review · Reviewer_xRrD · 2025-10-31

**Soundness:** 3
**Presentation:** 2
**Contribution:** 2
**Rating:** 2
**Confidence:** 3

**Summary:**

The paper benchmarks these modes on node classification tasks across homophilic vs. heterophilic graphs, short- vs. long-text datasets, and different model scales (from LLaMA to GPT-5). The authors report that Graph-as-Code consistently outperforms prompting and tool-calling due to better token efficiency and compositional reasoning.

**Strengths:**

- Timely and high-impact topic.
- Reproducibility transparency.

**Weaknesses:**

- Conceptual overlap between GraphTool and Graph-as-Code.
- Weak theoretical framing.

**Questions:**

1. The Graph-as-Code paradigm is described abstractly. If code generation and execution are treated as a form of tool invocation, then Graph-as-Code is effectively a special case of GraphTool with compositional or batched tool calls. Both rely on external execution environments to retrieve graph information.
2. In Prompting, neighborhood hops are truncated due to token limits. In GraphTool/Graph-as-Code, there is no equivalent limit, and the LLM can query indefinitely. This makes the modes incomparable in resource exposure: Graph-as-Code effectively sees more information than Prompting. How many tokens are used per query on average?
3. The work attributes improvement to “graph reasoning,” but Graph-as-Code might simply exploit code-LLM priors (e.g., structural induction, algorithmic recall). There is no control experiment using the same LLM on shuffled or random adjacency matrices to test if improvements are truly structure-dependent.
4. In GraphTool, how many reasoning steps are allowed?
5. Could authors present an example of how Graph-as-Code behaves in practice?

---

> ### Author Response · Authors · 2025-11-18
>
> We sincerely thank the reviewer for their detailed evaluation and for recognizing the significance of this work.
>
> > W1: Conceptual overlap between GraphTool and Graph-as-Code.
>
> > Q1: The Graph-as-Code paradigm is described abstractly. If code generation and execution are treated as a form of tool invocation, then Graph-as-Code is effectively a special case of GraphTool with compositional or batched tool calls. Both rely on external execution environments to retrieve graph information.
>
> **Answer:** This is an excellent point. We would like to provide some clarifications. All ReAct-style methods, including GraphTool, GraphTool+, and Graph-as-Code, are indeed a form of tool calls, which we evaluate against one another to assess their strengths and weaknesses. However, the core difference, and a key axis of variability in our study, is the level of flexibility and compositionality of the action space.
>
> GraphTool operates on a fixed, predefined set of simple APIs. Graph-as-Code provides the LLM with a full programmatic interface (a Pandas DataFrame) and a code interpreter. This allows the LLM to generate and execute arbitrary, compositional programs in a single turn.
>
> **This increased agency is the focus of our evaluation.** It is an open question in graph machine learning what level of abstraction and action space flexibility an LLM can best leverage. Our work provides the first clear evidence on this. For example, Finding 2 ("In ReAct-based methods, more flexible variants perform better on homophilic datasets" ) directly supports the idea that this increased compositional power is a significant advantage.

---

> ### Author Response · Authors · 2025-11-18
>
> > Q2: In Prompting, neighborhood hops are truncated due to token limits. In GraphTool/Graph-as-Code, there is no equivalent limit, and the LLM can query indefinitely. This makes the modes incomparable in resource exposure: Graph-as-Code effectively sees more information than Prompting.
>
> **Answer:** Thank you for raising this point. We would like to clarify that we are not proposing new paradigms; rather, our main aim has been to systematically compare these three strategies, Prompting, Tool-Use (which we refer to as GraphTool), and Code-Generation (which we term Graph-as-Code), and evaluate their respective benefits and drawbacks for researchers and practitioners.
>
> Thus, the varying resource exposure is not a confounding variable but rather a fundamental, defining property that differentiates these interaction modes. **We therefore intentionally compare these modes as they are used in practice**. The fact that Prompting is severely constrained by token limits is one of its primary, real-world weaknesses.
>
> To address your valid concern more directly, **we conducted a new experiment introducing a new mode: the 3-hop summary prompt**. This mode uses an iterative and recursive summarization process to compress information from distant hops, allowing us to include it without exceeding the token limit. The process begins by first going to each 2-hop node and using the LLM to summarize its 1-hop neighborhood (i.e., the 3-hop nodes). We then move to the 1-hop nodes and summarize their 1-hop neighborhood (the 2-hop nodes), but now each 2-hop node is represented by the 3-hop summary we just created. The final prompt for the target node then includes these richly summarized 1-hop neighbors, which now carry compressed 2-hop and 3-hop information. This allows the model to access 3-hop information that we could not have included before due to token limits.
>
> We also introduced a 2-hop and a 4-hop summary prompt that repeat the same logic. These experiments also allows for a fair comparison of resource exposure, as our logs confirm that our ReAct-based modes do not query beyond 4-hops over the below datasets. Here are the results comparing the new mode to the relevant modes from Table 3:
>
> | Method | citeseer | reddit | computer | photo | instagram | wiki-cs |
> | :--- | :--- | :--- | :--- | :--- | :--- | :--- |
> | 0-hop prompt | 68.20 | 47.90 | 65.50 | 69.80 | 48.00 | 74.00 |
> | 1-hop prompt | 68.30 | 59.30 | 86.10 | 85.80 | 56.10 | Limit |
> | 2-hop prompt | 69.40 | Limit | Limit | Limit | Limit | Limit |
> | 2-hop budget prompt | 70.20 | 54.40 | 86.00 | 85.60 | 54.50 | 80.80 |
> | **2-hop summary prompt** | 70.40 | 57.60 | 86.20 | 86.00 | 55.70 | 81.70 |
> | **3-hop summary prompt** | 69.80 | 57.00 | 85.80 | 85.60 | 55.80 | 81.00 |
> | **4-hop summary prompt** | 69.60 | 55.90 | 86.00 | 85.40 | 55.60 | 80.90 |
> | GraphTool | 68.30 | 56.25 | 80.80 | 77.00 | 47.80 | 76.27 |
> | Graph-as-Code | 71.80 | 61.60 | 86.20 | 86.40 | 56.40 | 82.20 |
>
> These new results provide several key insights. First, **the 2-hop summary prompt consistently outperforms both the 1-hop prompt and the 2-hop budget prompt**, establishing it as a much stronger and more token-efficient prompting baseline.
>
> Second, we observe a clear trend of diminishing, and even negative, returns as we extend this summarization to more distant hops. Performance peaks at the 2-hop summary and then slightly declines with the 3-hop summary and 4-hop summary. **Since we know adding hops is beneficial in these homophilic datasets, this decline indicates that the iterative summarization process is lossy.** Each summary of a summary compounds the information loss, eventually outweighing the benefit of including more distant nodes.
>
> Finally, even **these stronger, multi-hop summary baselines are still outperformed by Graph-as-Code**. This reinforces our Finding 4: "Graph-as-Code significantly outperforms Prompting and GraphTool on long-text datasets". This is expected as Graph-as-Code is not just seeing "more" information; it is adaptively and surgically retrieving the right information. It can, for example, retrieve only the labels of 2-hop neighbors (which are token-cheap) while ignoring their less informative features, or vice-versa. Prompting, even with iterative summarization, remains a non-adaptive, pre-computed information dump.
>
> We thus conclude that Graph-as-Code's programmatic, iterative approach allows it to execute a more flexible and efficient reasoning strategy, leading to its superior performance.

---

> ### Author Response · Authors · 2025-11-18
>
> > How many tokens are used per query on average?
>
> **Answer:** We provide the average token counts for queries on our long-text benchmarks. **These token counts demonstrate why the token limit is such a critical issue:**
>
> | Dataset | 0-hop | 1-hop | 2-hop | GraphTool | Graph-as-Code
> | :--- | :--- | :--- | :--- |:--- |:--- |
> | reddit | 1,762 | 10,300 | 42,100 | 16,100 | 14,900 |
> | photo | 3,000 | 36,100 | 104,700 | 43,300 | 41,500 |
> | wiki-cs | 30,200 | 202,500 | 1,157,000| 61,700 | 57,100|
>
> As the table shows, the total token count increases significantly from `reddit` to `photo`, and again to `wiki-cs`. This trend is expected, as it mirrors the increase in the average text length and average node degree for these datasets. Consequently, **1-hop and 2-hop prompts quickly exceed the context window of 200,000 tokens for our default o4-mini model forcing truncation.** In contrast, GraphTool and Graph-as-Code use much smaller token counts, as they only retrieve specific pieces of information (e.g., a list of neighbor IDs or a single node's features) and build context iteratively.
>
> > Q3: The work attributes improvement to “graph reasoning,” but Graph-as-Code might simply exploit code-LLM priors (e.g., structural induction, algorithmic recall). There is no control experiment using the same LLM on shuffled or random adjacency matrices to test if improvements are truly structure-dependent.
>
> **Answer:** This is a great point. To isolate the effect of graph structure, we repeated our experiments across 7 datasets, replacing the true adjacency matrix with a randomly shuffled one. This process preserves all node features and the graph's degree distribution but destroys all meaningful structural patterns. The results, averaged over 10 different random adjacency matrices, are as follows:
>
> | | cora | arxiv | cornell | texas | citeseer | photo | wiki-cs|
> | :--- | :--- | :--- | :--- |:--- |:--- |:--- |:--- |
> | Random | 14.13 | 2.44 | 21.74 | 8.40 | 16.80 | 8.20 | 9.90 |
> | Label propagation| 13.29 | 3.85 | 30.48 | 10.34 | 13.10 | 8.40 | 9.60 |
> | 0-hop prompt | 64.21 | 68.10 | 81.57 | 53.20 | 68.20 | 69.80| 74.00|
> | 1-hop prompt | 58.10 | 61.50 | 74.30 | 47.10 | 60.30 | 63.20| Limit|
> | 2-hop prompt | 44.50 | 47.20 | 60.10 | Limit | 45.10 | Limit| Limit|
> | GraphTool | 56.20 | 59.80 | 72.40 | 45.90 | 58.10 | 61.10| 66.20|
> | Graph-as-Code | 56.50 | 59.10 | 72.90 | 45.20 | 58.80 | 60.90| 66.70|
>
> As the results show, **the performance of all structure-aware methods significantly decrease** when compared with their original results (Tables 1-3). This **confirms they are attempting to use the neighborhood information**, which now only adds noise. These additional trends arise:
>
> **(1) The 0-hop prompt's performance is unaffected**, since it never queries the neighborhood. It is thus immune to the structural noise and becomes the best-performing baseline.
>
> **(2) Label Propagation is on par with a random guess**. This is expected, as it relies exclusively on the now-meaningless structural information.
>
> **(3) Performance degradation scales with the amount of noise introduced.** 2-hop prompting is impacted more severely compared to the 1-hop prompting, as it ingests the largest volume of noisy neighborhood context.
>
> **(4) GraphTool and Graph-as-Code prove more robust than the naive 2-hop prompt.** This suggests that while the LLM's reasoning is confused by the noise, it can still mitigate the noise better than a simple prompt expansion, likely by weighing its own (correct) features more heavily than its neighbors' (noisy) information.
>
> These results strongly indicate that the superior performance of Graph-as-Code and other interaction modes seen in our main paper is not an artifact of code-generation priors, but a direct result of effectively utilizing the true graph structure. **This experiment also reinforces Finding 6:** "Graph-as-Code is more robust than Prompting." While 100% random noise hurts all methods, Graph-as-Code's performance degrades less severely than the 2-hop prompt, demonstrating a superior ability to handle noisy or irrelevant structural information, even in this extreme scenario.

---

> ### Author Response · Authors · 2025-11-18
>
> > Q4: In GraphTool, how many reasoning steps are allowed?
>
> **Answer:** We set an upper bound of 50 think-act-observe steps to ensure the agent was not prematurely terminated. In practice, no experiment reached this limit. The models typically converged on a prediction within 10 steps, depending on the dataset and the specific variant (GraphTool vs. GraphTool+).
>
> > Q5: Could authors present an example of how Graph-as-Code behaves in practice?
>
> **Answer:** Yes, below is a representative example log from Graph-as-Code on a widely used dataset (cora). The reasoning pattern shown here appeared frequently in our runs and reflects typical behavior on this dataset.
>
>
> **Task:** You are solving a node-based reasoning task for node 133. You have a pandas DataFrame `df` where each row corresponds to a node, indexed by its `node_id`.
>
> **Instructions:** Always begin with reasoning. After your reasoning, provide a single, valid pandas command on a new line.
>
> **Schema structure:**
>  - The DataFrame index is the node id. Access a row by node id with: `df.loc[node_id]`.
>  - The column `features` stores each node’s textual description: `df.loc[node_id, 'features']`.
>  - The column `neighbors` stores a list of neighbor node IDs: `df.loc[node_id, 'neighbors']`.
>  - The column `label` contains the integer node label if it belongs to the training set; otherwise `None`.
>
> **Available class labels:**
>
> 0: Rule Learning,
>
> 1: Neural Networks,
>
> 2: Case Based,
>
> 3: Genetic Algorithms,
>
> 4: Theory,
>
> 5: Reinforcement Learning,
>
> 6: Probabilistic Methods
>
> **Request 1:** I need to classify node 133. I will start by inspecting its features and its label.
>
> `df.loc[133][['features', 'label']]`
>
> **Response 1:** `features: 'A Reinforcement Learning Approach to Job-Shop Scheduling'`
> `label: None`
>
> **Request 2:** The label is unknown. The features strongly suggest 'Reinforcement Learning' (class 5). I will check the labels of its neighbors to confirm this.
>
> `df.loc[df.loc[133, 'neighbors'], 'label']`
>
> **Response 2:**
>
> `node_id label`
>
> `134      5.0`
>
> `135      5.0`
>
> `707      5.0`
>
> `1588     5.0`
>
> `1589     NaN`
>
> `2048     1.0`
>
> **Request 3:** I have 5 neighbors with known labels. Four are class 5 ('Reinforcement Learning') and one is class 1 ('Neural Networks'). The majority class among neighbors is 5. The presence of class 1 ('Neural Networks') also supports this, as reinforcement learning is highly correlated with neural networks. This neighborhood evidence strongly matches my initial hypothesis from the node's own features. I am confident in this prediction.
>
> `Answer: 5`
>
> **Thank you for this suggestion. This log illustrates the practical, step-by-step reasoning process of the Graph-as-Code mode.** It shows the agent forming a hypothesis from its own features (Class 5, 'Reinforcement Learning') and then programmatically testing that hypothesis against its neighborhood -- even noting the correlation between Class 1 and Class 5. This demonstrates a more complex reasoning process than simple neighborhood voting.
>
> We agree that these concrete examples are valuable for illustrating the qualitative differences between the interaction modes. **In the camera-ready version, we will add a dedicated section to the appendix with this example and others** to provide a clear, side-by-side comparison of the Prompting, GraphTool, and Graph-as-Code modes.
>
> **We will include all the experimental results and suggestions provided by the Reviewer in the final version with the appropriate discussion.**
>
> We hope our responses address your concerns, and thank you for helping us improve the paper. We would greatly appreciate your reconsideration of the evaluation and increasing your score.

---

> ### Author Response · Authors · 2025-11-24
>
> **Thank you for encouraging the resource exposure experiment, which led us to derive the summary prompt mode**, and for encouraging the random adjacency matrices experiment. We believe to have addressed all of your concerns and **would highly appreciate your response** as this would give us a chance to address any remaining issues. Thank you.

---

> > ### Comment · Reviewer_xRrD · 2025-11-25
> > **Overall good**
> >
> > Thanks to the authors for the clarifications. Most of my earlier concerns have been addressed. While I would still have liked to see more graph-specific insights beyond demonstrating that CodeAct improves performance (a finding already well-established in many domains). The systematic exploration across interaction modes, graph regimes, and data characteristics is valuable, and applying CodeAct in the graph domain represents a meaningful empirical contribution. Overall, this paper is a good attempt in this direction, and I am willing to raise my score.

---

> ### Author Response · Authors · 2025-11-28
>
> We are glad that our rebuttal addressed your concerns, and **we sincerely thank you for raising your score to 6.**

---

### Author Response · Authors · 2025-12-01
**Summary of rebuttal and changes**

We wish to express our sincere gratitude for the additional time and effort you are dedicating to evaluating our work. To facilitate your assessment, we provide below a concise overview of the main changes introduced during the rebuttal:

* **Reviewer xRrD (Score 2 → 6):** Conducted an experiment over random adjacency matrices to prove Graph-as-Code reasons over the structure; introduced a "summary prompt" mode to address the difference in resource exposure across modes where prompting has limiting k-hop information exposure. The reviewer raised their score to 6, acknowledging that the "systematic exploration... is valuable."

* **Reviewer 37Ma (Score 6 → 8):** Added thorough comparisons with GNN baselines and Hybrid LLM+GNN models as requested. The reviewer raised their score to 8, noting the paper is "an insight work."

* **Reviewer RTGA (Score 8):** Added token usage data to help assess the performance-efficiency trade-offs of the different modes; introduced a new synthetic "Shortest Path" experiment to include other graph computational problems. The reviewer replied that the response "thoroughly addressed my concerns" and that they would "champion this paper with greater confidence."

* **Reviewer PyZw (Score 6 → 8):** Added thorough comparisons with GNN baselines and Hybrid LLM+GNN models as requested; added token counts and latency metrics to help assess the performance-efficiency trade-offs of the different modes. The reviewer raised their score to 8, stating that "all concerns were addressed."

These changes led to **scores of 8, 8, 8, and 6**, corresponding to an **average of 7.5**, prior to the review period being frozen. All reviewers recognized that our large-scale study spans multiple previously unexplored axes of variability and offers new insights into LLM capabilities on graphs. **We sincerely hope these insights will benefit researchers, practitioners, and the wider graph machine learning community alike in designing the next generation of LLM-based graph methods.**

We trust that this summary will be beneficial to your assessment.

---

### Meta-Review · Area_Chair_Sh2a · 2026-01-06

**Summary:**

The decision to accept this paper is grounded in its systematic, large-scale empirical study of LLM graph reasoning and the proposal of the novel "Graph-as-Code" interaction paradigm. Reviewers highly valued the rigor and breadth of the experimental design, spanning six key dimensions including interaction modes and model scales, and confirmed that the proposed approach significantly outperforms traditional prompting and tool-based methods, particularly in tasks involving long texts and high-order graphs. Furthermore, the authors effectively addressed the reviewers' specific concerns during the rebuttal phase, solidifying the decision.

**Reviewer Concerns:**

Additionally, the authors have effectively addressed numerous reviewer concerns during the rebuttal phase. Therefore, the decision is to Accept.

**Reviewer Scores:**

6，8，8，8

---

### Decision · Program_Chairs · 2026-01-26

Accept (Oral)